# Decoding brain states on the intrinsic manifold of human brain dynamics across wakefulness and sleep

Joan Rué-Queralt [1✉], Angus Stevner[2,3], Enzo Tagliazucchi[4], Helmut Laufs[5,6], Morten L. Kringelbach [2,3], Gustavo Deco [1,7,8,9] & Selen Atasoy[2,3]

Current state-of-the-art functional magnetic resonance imaging (fMRI) offers remarkable imaging quality and resolution, yet, the intrinsic dimensionality of brain dynamics in different states (wakefulness, light and deep sleep) remains unknown. Here we present a method to reveal the low dimensional intrinsic manifold underlying human brain dynamics, which is invariant of the high dimensional spatio-temporal representation of the neuroimaging technology. By applying this intrinsic manifold framework to fMRI data acquired in wakefulness and sleep, we reveal the nonlinear differences between wakefulness and three different sleep stages, and successfully decode these different brain states with a mean accuracy across participants of 96%. Remarkably, a further group analysis shows that the intrinsic manifolds of all participants share a common topology. Overall, our results reveal the intrinsic manifold underlying the spatiotemporal dynamics of brain activity and demonstrate how this manifold enables the decoding of different brain states such as wakefulness and various sleep stages.

[1] Center of Brain and Cognition, Universitat Pompeu Fabra, Barcelona, Spain. [2] Centre for Eudaimonia and Human Flourishing, University of Oxford, Oxford, UK. [3] Center for Music in the Brain, Aarhus University, Aarhus, Denmark. [4] Instituto de Física de Buenos Aires and Physics Deparment (University of Buenos Aires), Buenos Aires, Argentina. [5] Department of Neurology and Brain Imaging Center, Goethe University, Frankfurt am Main, Germany. [6] Department of Neurology, University Hospital Schleswig-Holstein, Christian-Albrechts-University, Kiel, Germany. [7] Institució Catalana de Recerca i Estudis Avancats (ICREA), Barcelona, Spain. [8] Department of Neuropsychology, Max Planck Institute for Human Cognitive and Brain Sciences, Leipzig, Germany. [9] School of Psychological Sciences, Monash University, Melbourne, Australia. ✉email: joan.rue.q@gmail.com

The brain is an immensely complex dynamical system capable of generating an extremely large repertoire of neural activity patterns. Despite such potential, whole-brain imaging of spontaneous brain activity is shown to exhibit highly constrained patterns[1,2]. Hence, in terms of temporal dynamics, it has been hypothesized that brain activity is governed by metastable dynamics[3–7] and lies on a low-dimensional smooth manifold embedded in the high-dimensional space of the neuroimaging data[8–10]. Supporting this hypothesis, growing experimental evidence demonstrated the presence of continuous attractors in the brain[11–14]. Indeed, the number of degrees of variability of the recorded neural activity during perception and memory[11], learning[15], and motor tasks[13] is shown to be substantially lower than the dimensionality of the space defined by the number of recorded neurons. Even during task-free conditions, such as sleep and resting wakefulness, populations of neurons show very structured dynamics, which can be represented as 1D ring-shaped manifolds[12].

The nature of the manifold underlying large-scale dynamical processes in the human brain can be explored at the whole-brain level using functional neuroimaging data, such as fMRI or magnetoencephalography (MEG). Consider, for instance, a series of fMRI acquisitions, which yield an $\mathbb{R}^\alpha$ image for each time point, where $\alpha \sim 10^7$ is the number of voxels (even though these voxels are not completely independent due to the spatial smoothness of fMRI[16]). Interestingly, the range of possible configurations of brain activity derived from this image does not span this high-dimensional state-space, but is rather limited to a subspace[10]. In this work, we hypothesized that due to the anatomical and physiological constraints, and most importantly, due to strong correlations among neural populations[17], not only does the large-scale dynamics of human brain activity span a lower-dimensional subspace but it actually lies on a smooth manifold. Here, besides testing this hypothesis we also investigate whether this compact manifold representation can be utilized to characterize different dynamical regimes in the space of all brain states, in particular to characterize different stages of the human sleep cycle. To this end, we developed a framework to estimate the intrinsic low-dimensional manifold underlying the brain dynamics as measured by the fMRI data.

A normal human sleep cycle is divided into wakefulness, rapid-eye-movement (REM) sleep, and three stages of non-REM (NREM) sleep[18]. While wakefulness and REM sleep are both characterized by low amplitude and high-frequency electroencephalography (EEG) signals, the three NREM sleep stages are defined by a gradual 'slowing down' of the EEG oscillations. The decrease in arousal is mapped onto a path from N1 (light sleep with increased amplitude of low-frequency EEG oscillations) over N2 (a deeper sleep stage than N1 that also includes sleep spindles and K-complexes) to N3 (slow-wave) sleep. Over the years this mapping has been refined and developed into polysomnography, the current gold standard description of sleep, where EEG is combined with recordings of eye movements, muscle tone, respiration, and heart rate to categorize sleep into stages.

While the polysomnography represents a good mapping between brain activity and arousal, the limited spatial resolution achieved through the readout from merely a few EEG electrodes offers little information about more fine-grained aspects of brain activity during sleep. Over the past couple of decades, other functional neuroimaging techniques such as positron emission tomography (PET) and fMRI have been also utilized to extract the whole-brain correlates of different sleep stages defined by polysomnography[19–21]. More recently, the framework of whole-brain functional connectivity (FC) and resting-state networks[22] have been explored to provide more comprehensive accounts of how the brain's large-scale functional architecture changes between wakefulness and particularly NREM sleep (for review, see ref. [23]). Through the combination of support vector machine (SVM) algorithms and resting-state networks found in fMRI with polysomnography-verified stages, it has been thoroughly demonstrated that sleep alters large-scale FC, to an extent where SVMs can be trained to perform reliable sleep staging solely based on FC information derived from the fMRI[24–26]. These, together with previous studies investigating sleep-related changes in resting-state networks[27–29] and graph-theoretical properties[30–32], rely on the linearity assumption of the data. As such, these techniques assume that relations between the time courses of two different voxels (in fMRI) or electrodes (in EEG) are linear; and hence neglect the nonlinear properties of brain activity, which have been suggested to be relevant for sleep processes: evidence from intracortical recordings have pointed out the differences in the occurrence of neuronal avalanches between wakefulness and sleep[33], and it has been demonstrated that the addition of nonlinear features of the data can improve the discrimination between sleep stages using scalp EEG[34].

Here we propose an algorithm to find the low-dimensional smooth manifold underlying fMRI BOLD activity during the human sleep cycle, which we call the intrinsic manifold of brain dynamics. To this end, we first estimate a state representation defined by the time-resolved FC matrix in phase space termed coherence connectivity dynamics (CCD). This matrix captures the synchrony characteristics of brain activity at a given time point by estimating the phase coherence among all pairs of brain regions[35]. We utilize a manifold learning technique known as Laplacian eigenmaps[36] to nonlinearly reduce the dimensionality of the phase coherence space. Laplacian eigenmaps accounts for nonlinear relationships between individual datapoints (in our case between the CCD of two different fMRI time points). Furthermore, being a local manifold learning approach based on the eigenfunctions of the Laplace operator, Laplacian eigenmaps is computationally efficient to be applied to large datasets[36,37] and has been successfully applied to extract the manifold structure underlying neural activity from MEG and EEG time series[38]. In this work, we utilize the Laplacian eigenfunctions to extract the relevant information from brain dynamics in the temporal domain. Remarkably, when applied to the spatial domain, specifically to the structural connectivity of the human brain, i.e. the human connectome, Laplacian eigenfunctions yield the connectome harmonics[39]. Connectome harmonics were shown to reveal the functional networks of the human brain. Furthermore, when applied to the cortical structure of the brain, Laplacian eigenfunctions yield the cortical eigenmodes, which capture the spatiotemporal patterns of distinct sleep-states[40].

In this work, we apply Laplacian eigenmaps to the temporal dimension of fMRI data in order to reveal the intrinsic manifold of brain dynamics while also taking advantage of the high spatial resolution of fMRI data. We acquired recordings of brain activity simultaneously with fMRI and polysomnography (including EEG) from 18 participants during wakefulness and sleep (data acquisition originally published in ref. [24]). From these recordings, we used only the fMRI data to estimate the intrinsic manifold of brain activity, while utilizing the expert sleep scoring of the simultaneously acquired EEG recordings to evaluate sleep stage classification on the intrinsic manifold representation. A simple linear SVM, when applied to the estimated intrinsic manifold representation, yielded a classification accuracy of 96% and significantly outperformed linear dimensionality reduction methods such as principal component analysis (PCA). These results not only reveal the low-dimensional manifold underlying the complex brain dynamics but also demonstrate the intrinsically nonlinear nature of the differences in these spatiotemporal patterns, in particular between wakefulness and different sleep stages.

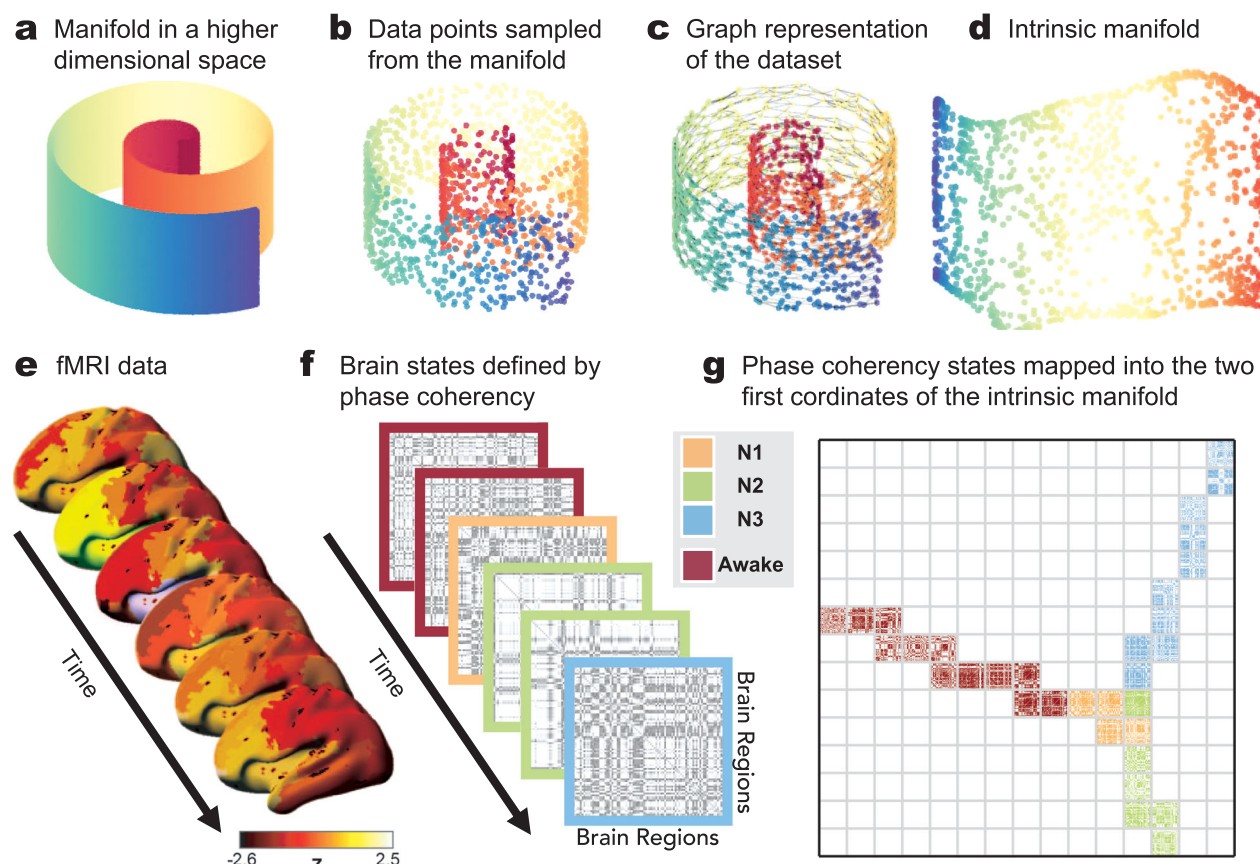

**Fig. 1 Intrinsic manifold framework. a–d** A classic example that illustrates manifold embedding; i.e., manifold learning applied to the swiss roll data, which is intrinsically a two-dimensional dataset yet represented in a higher (three-dimensional) space. To estimate the low dimensional embedding of the sampled dataset (**b**), we first create a graph representation (**c**), where the nodes represent the data points shown in (**b**), which are sampled from the underlying manifold illustrated in (**a**), and the edges indicate the relations (distances and/or similarities) between data points. **d** The manifold is embedded into the low-dimensional representation that matches its intrinsic dimensionality using the Laplacian eigenmaps manifold learning. **e–g** Our framework applying the same manifold learning approach, i.e. Laplacian eigenmaps, to extract the manifold underlying brain dynamics measured in fMRI data. **e** For each time point, the fMRI BOLD signal is parcellated into the 90 brain areas defined by the AAL template and pre-processed as explained in the "Methods" section. **f** Using the parcellated fMRI data, the instantaneous phase is computed via Hilbert transform and the phase coherence among brain areas is estimated. This phase coherency matrix characterizes the pairwise synchrony relations between each pair of brain areas at any given time point. **g** The intrinsic manifold (here illustrated as two-dimensional) underlying the set of all instantaneous phase coherence states is estimated using the Laplacian eigenmaps method. To visualize the changes in phase coherency throughout the intrinsic manifold, for illustration purposes we defined 2-dimensional (2D) bins using the two manifold dimensions, and computed the average phase coherency of data points in those bins. Different colors indicate different sleep stages and wakefulness as defined by polysomnography.

## Results

**Revealing the intrinsic manifold of brain dynamics**. In order to test whether the dynamics of brain activity lie on a smooth, low-dimensional manifold (exemplified in Fig. 1a–d), as previously hypothesized[41], we introduce a new method to estimate the intrinsic manifold underlying human fMRI data. We first extract the instantaneous phase signal of the ultraslow fluctuations (0.04–0.07 Hz) of the fMRI BOLD activity and compute the phase coherence metric for each participant[42] (Fig. 1e, f). Phase coherence is a commonly used estimator of the instantaneous synchronization between all pairs of brain regions, which are defined through automated anatomical labelling (AAL)[43] in this study. An important feature of phase coherence connectivity estimates is that they capture complex dynamics within the data, they are robust to inter-subject variability, and do not require of any temporal windowing[42]. From phase coherence, we then obtain the CCD matrix[35], the matrix containing the temporal relationships of these spatial synchrony patterns. The estimated CCD matrix defines the feature space, which describes brain dynamics by capturing all temporal relations between brain states

occurring in the fMRI data. In order to compute the intrinsic manifold, we nonlinearly embed the CCD feature space to a lower-dimensional space. The embedding consists of two main steps: first, a graph representation of the nonlinear relations within the data is formed by pruning the weak connections (similarities) between the spatial synchrony patterns. Commonly, such pruning of the similarity matrix is preformed through thresholding or $k$-nearest-neighbour selection[36]. However, here we use a method that is more robust to sampling inhomogeneities, the relaxed minimum spanning tree[44] (RMST, see the "Methods" section). Once the graph representation is created, the second step of the embedding consists of obtaining the eigenfunctions of the graph Laplacian. This embedding method is commonly known as the Laplacian eigenmaps method[36]. The eigenfunctions with the smallest nonzero eigenvalues form the basis of the new embedding (Fig. 1g). By applying this method to fMRI data of 18 participants acquired in wakefulness and sleep, we reveal each subject's intrinsic manifold underlying their brain dynamics. For each individual subject, the intrinsic manifold yields a smooth, continuous representation of the brain dynamics

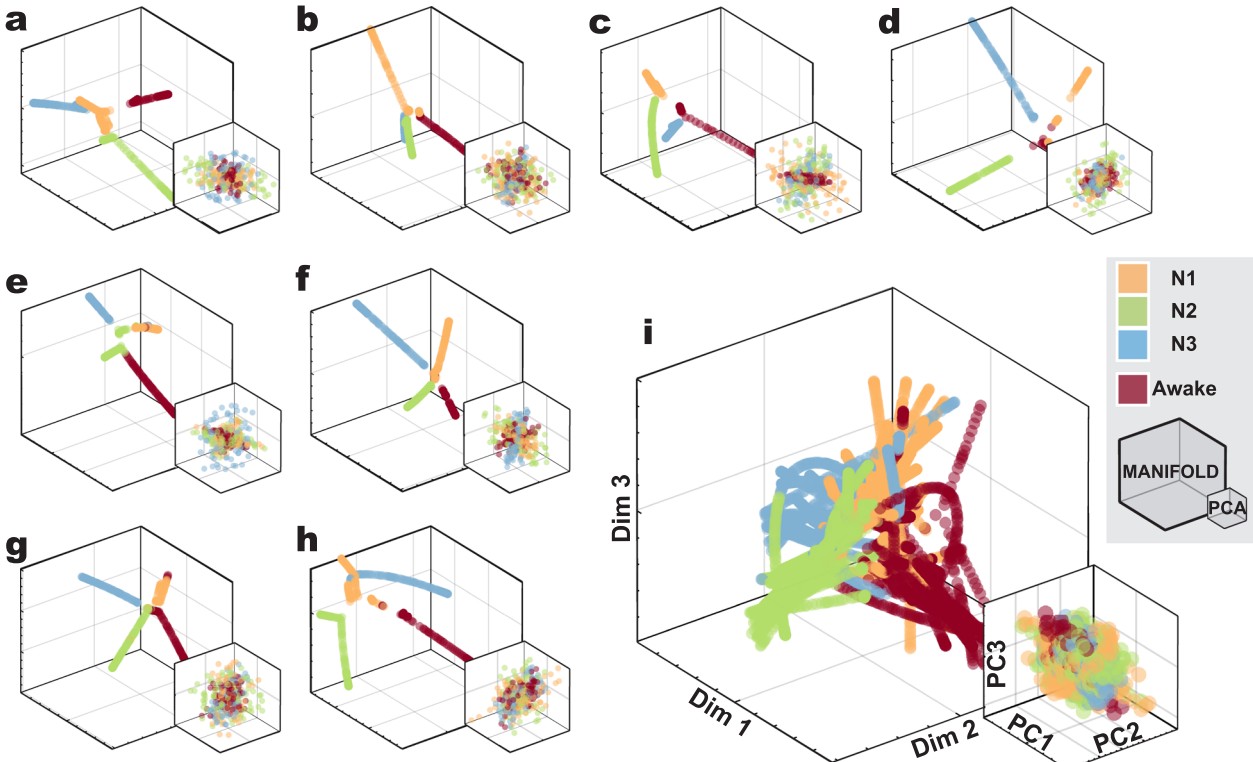

**Fig. 2 Representation of the brain activity fMRI BOLD data during wakefulness and sleep embedded in lower-dimensional spaces.** The plots show the data embedded into the three first dimensions of the intrinsic manifold (large coordinate system) and into the three principal components derived from PCA (small coordinate system). **a–h** Each separate coordinate system corresponds to the data of eight different participants, embedded individually. **i** Intrinsic manifolds from all 18 participants, aligned jointly into the group manifold. For all cases, nonlinear embedding of the data into their intrinsic manifold led to well-structured intrinsic manifolds with a clearer separation of different sleep stages (as defined through polysomnography) compared to the linear embedding given by PCA.

that is implicitly low dimensional ($d = 7$ in our study) as shown in Fig. 2 (for eight representative subjects, see Supplementary Fig. 1 for all subjects' data). The word continuity, is used here in the spatial sense referring to the structured and non-random spatial distribution of data points on the intrinsic manifold and does not refer to the temporal smoothness, i.e., temporal continuity of the data points on the manifold. In this low dimensional representation, time-points acquired during the same stage (sleep stages and wakefulness) lie together, yet different stages fall onto different branches and are also separated through shortcuts in the temporal dynamics, where the brain dynamics exhibit a jump on the intrinsic manifold from one branch (corresponding to a particular sleep stage, e.g., N3) to that of another (e.g., awake). This effect is shown in detail in Supplementary Fig. 2 and Supplementary Movie 1. From these results, one can appreciate the effectiveness of the presented intrinsic manifold method to separate different brain states such as different stages of sleep and wakefulness.

**Decoding brain states in the intrinsic manifold of human brain dynamics.** Following the rationale that different sleep stages are associated with characteristically different brain states, we hypothesized that they can be accurately decoded using the intrinsic manifold underlying the brain dynamics. In order to test this hypothesis, we trained a linear SVM classifier on the intrinsic manifold of brain dynamics and estimated the discriminative power of the classifier for each of the sleep stages as well as wakefulness (see the "Methods" section). For individual subject analysis, we utilized a multi-class 10-fold cross-validation

approach, using 9/10 of the whole data as training and the remaining 1/10 as test set in order to test the discriminative power of our approach for each of the three sleep stages and wakefulness (i.e., one-vs-all comparisons). Figure 3 shows the accuracy for each possible stage in low-dimensional spaces ($d = 7$, see Supplementary Fig. 3 for accuracy in $d = 3$). For each stage, we found significantly high classification accuracy ($p$-value < 0.001, Monte-Carlo phase randomized simulations, corrected for multiple comparisons via FDR, see Supplementary Tables 1–3 and see the subsection "Statistical significance analysis" in the "Methods" section) with average accuracy being 96 ± 4%. We also decoded brain activity in low-dimensional spaces using a 1-vs-1 SVM approach (explained in the "Methods" section). For each stage-to-stage comparison, by representing the data in their intrinsic manifolds we found significantly high decoding accuracy ($p$-value < 0.001, Monte-Carlo simulations, corrected for multiple comparisons via FDR, see Supplementary Tables 1–3 and see sunsection "Statistical significance analysis" in the "Methods" section) with average accuracy being 99 ± 3%. These results demonstrate that the brain activity associated with different brain states such as different sleep stages and wakefulness becomes highly separable on intrinsic manifold of brain dynamics and hence can be robustly decoded in this low-dimensional manifold representation.

To determine the intrinsic dimensionality of the data, we performed the sleep-stage classification for all possible dimensionalities (see Fig. 3l, n). Our results indicate that from $d = 7$ up to the original dimensionality (90), the addition of dimensions in the intrinsic manifold does not improve the decoding accuracy (for all dimensions $d > 7$, $p$-value > 0.05, Wilcoxon Rank-sum

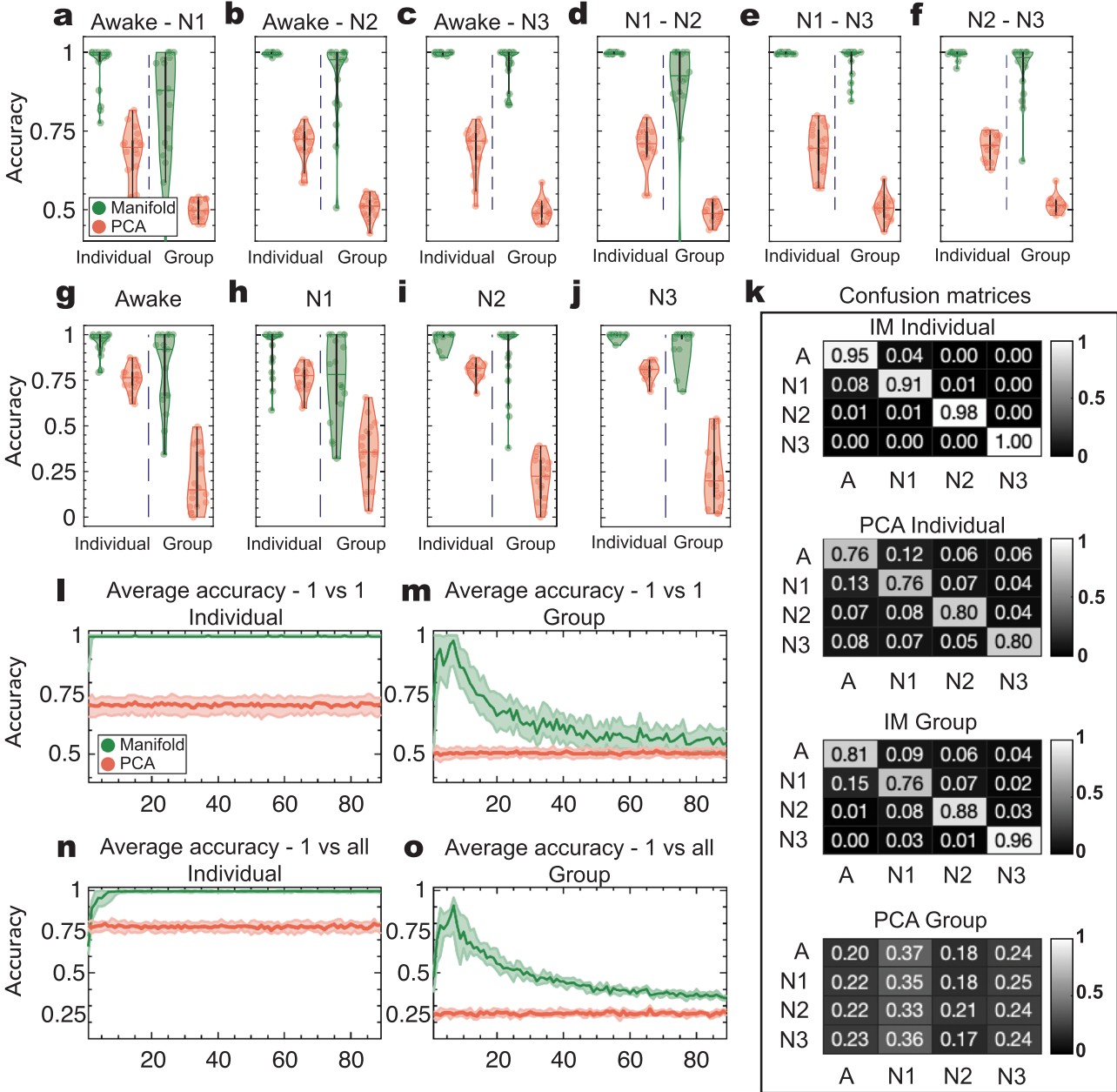

**Fig. 3 Accuracy of brain state decoding on the intrinsic manifold of brain dynamics and on PCA.** Decoding accuracies in the feature space defined by the intrinsic manifold (IM) and PCA space for different experiments. **a–f** The accuracies of the SVM 1-vs-1 classification between **a** wakefulness and N1, **b** wakefulness and N2, **c** wakefulness and N3, **d** N1 and N2, **e** N1 and N3, and **f** N2 and N3. **g–j** The accuracies of the SVM 1-vs-all classification for each stage: **g** wakefulness, **h** N1, **i** N2 and **j** N3. The accuracy is defined as the ratio between the number of true positives and the total number of tested time points. The boxplots' centrality is indicated by the median, and the boxes extend between 25th and 75th percentiles. Each colored circle corresponds to the classification accuracy for each single subject (in the case of individual analysis, left of the dashed line) and to the accuracy of each leave-one-subject-out round (in the case of group analysis, right to the dashed line). The classification accuracies on the intrinsic manifold and in PCA space are represented by green and red dots, respectively. Classifications are performed in spaces of dimensionality $d = 7$ (see Supplementary Fig. 3 for $d = 3$). For all classifications, intrinsic manifold classification yields significantly higher accuracies (for all comparisons, $p$-value < 0.001, Wilcoxon Rank-sum two-sided test, corrected for multiple comparisons via FDR). **k** Confusion matrices obtained from the 1-vs-all classification experiments (shown in **g–j**. **l**, **m** show the average accuracy across all stage-to-stage (1-vs-1) classifications for varying dimensionality of the embedding spaces for individual participants (**l**) and for group analysis (**m**), respectively. **n**, **o** show the average accuracy for all stages (1-vs-all) classifications for varying dimensionality of the embedding spaces for individual participants (**n**) and for group analysis (**o**), respectively. The solid lines indicate the median of the distribution across classifications and shaded areas indicate 25th and 75th percentiles.

two-sided test, corrected for multiple comparisons via FDR). Thus, the intrinsic dimensionality of the manifold can be estimated to be $d = 7$.

These results demonstrate that the intrinsic manifold reveals the hidden topology underlying brain dynamics, where different brain states such as distinct sleep stages and wakefulness become highly separable in a low-dimensional manifold. Moreover, by robustly decoding different brain states on the intrinsic manifold, our results strongly suggest that the brain dynamics governing these different brain states exhibit significantly different characteristics.

**Common intrinsic manifold topology across participants**. Next, we evaluated the consistency of the topology of the intrinsic manifold of brain dynamics across participants. To this end, we aligned the intrinsic manifolds of different subjects (see Fig. 2i and see the "Methods" section) and used the leave-one-subject-out approach: we trained a linear SVM classifier on the aligned intrinsic manifolds generated with data from all participants except for one and performed inference on the left-out subject (see the "Methods" section). This group analysis yielded significantly high classification accuracy with the average accuracy at the group level being $85 \pm 9\%$ ($p$-value < 0.001, Monte-Carlo phase randomized simulations, corrected for multiple comparisons via FDR, see Supplementary Tables 1–3 and see the subsection "Statistical significance analysis" in the "Methods" section).

In particular, we found that the sleep stages N2 and N3 had the highest separability on the intrinsic manifold, with an average classification accuracy of $99 \pm 3\%$ ($92 \pm 15\%$ for group analysis) (Fig. 3). Comparisons involving the awake and N1 stages were the least separable, yet still leading to a significantly higher classification accuracy compared to chance (average accuracies $93 \pm 10\%$ and $78 \pm 22\%$, respectively, $p$-values < 0.001, Monte-Carlo phase randomized simulations, corrected for multiple comparisons via FDR). The confusion matrices (see Fig. 3k) show that the time-points belonging to the awake and N1 stages are commonly classified as part of the same class.

These findings indicate that the feature space described by the CCD matrix encodes crucial information about brain dynamics in sleep, which is generalizable across participants, and that the intrinsic manifolds successfully reveal the characteristic structure underlying different brain states with the small number of degrees of the variability of the data; i.e., its intrinsic dimensionality (Fig. 3m and o show that classification accuracy is optimal for $d = 7$ in our study). Crucially, these results demonstrate that the intrinsic manifolds underlying brain dynamics of different participants share a common topology that is primarily constrained by the brain state (wakefulness and different sleep stages) and not the individual differences between participants.

**Nonlinear transformation of brain activity between different sleep stages**. The intrinsic manifold provides a low-dimensional representation of brain dynamics, which respects nonlinear relations between data points (in our case brain states occurring at different time instances). Unlike linear dimensionality reduction techniques such as PCA[45] or independent component analysis (ICA)[46], the intrinsic manifold is estimated by a nonlinear mapping from the high to the low-dimensional space and preserves the nonlinear metric properties of the high-dimensional data. In order to test whether these nonlinearities play a crucial role in the decoding of different brain states in sleep and wakefulness, we compared the accuracies of classifications performed on the intrinsic manifold and on PCA of brain dynamics. Figure 2 shows the first three dimensions of the individual intrinsic manifolds and PCA of eight different participants, as well as the aligned intrinsic manifolds from all participants gathered together (Supplementary Fig. 4 shows the first 6 dimensions). We observed that the linear embedding given by PCA performed poorly at capturing the structure underlying brain dynamics, yielding an average classification accuracy of $78 \pm 3\%$ in individual and $25 \pm 7\%$ in group analysis (25% being chance level, see Supplementary Tables 1 and 2). The reason why the intrinsic manifold provides a suitable embedding for sleep stage classification becomes evident by visual inspection of Supplementary Fig. 2, and Supplementary Movie 1. These show how temporal traces belonging to different sleep stages are clustered separately on the intrinsic manifold,

while the data points belonging to the same stage lie on the same branch of the manifold, even though the temporal structure of the data is disrupted, inducing discontinuities in the temporality of the used data. Using a nonparametric test, we assessed the statistical significance of the differences between the classification accuracies of PCA and intrinsic manifold (see the "Methods" section). All stages showed significantly higher classification accuracy on the intrinsic manifolds compared to PCA (for all comparisons, $p$-value < 0.001, Wilcoxon Rank-sum two-sided test, corrected for multiple comparisons via FDR). We cannot be certain whether PCA's lower decoding performance is attributed to its linear nature, or to other details of its framework. We can, however, be certain that nonlinearities play a crucial role in the data, as indicated by the significantly improved accuracies on the intrinsic manifolds with respect to the linear surrogate manifolds ($p$-value < 0.001, corrected for multiple comparisons via FDR, see Supplementary Table 1 and Statistical significance analysis in the "Methods" section). These results demonstrate that the nonlinear relations among data samples, which are captured by the intrinsic manifold, play a crucial role in the decoding and characterization of the different brain states occurring in wakefulness and in NREM sleep.

**Robustness of temporal harmonics for sleep stage classification**. So far, our results suggest that classification performed on the intrinsic manifold of brain dynamics measured with fMRI allows for an accurate decoding of the different sleep stages as well as wakefulness (stages being categorized with polysomnography in EEG). In order to assess the robustness of this brain state decoding performed on the intrinsic manifold, we assessed the receiver-operating characteristic (ROC) of the classification by varying the decision threshold used for the binary classification along each of the three dimensions separately. Based on signal detection theory, ROC is a commonly used analysis to validate the robustness of binary classifiers. ROC analysis provides a simple way to evaluate the trade-off between sensitivity and specificity of the classification. In the ideal scenario, where two classes are fully separable, a binary classifier would yield a performance approaching the top-left corner, where sensitivity and (1-specificity) are 1 and 0, respectively, leading to an area under the curve (AUC) value of 1. Figure 4 and Supplementary Table 2 demonstrate the ROC curves for all stage-to-stage comparisons for a single dimension for which the best performance was achieved. We observed an average AUC value of $0.98 \pm 0.02$ on the intrinsic manifold, while the AUC of the ROC curve of the classification performed on PCA was $0.51 \pm 0.01$ on average (for all comparisons, $p$-value < 0.001, Wilcoxon Rank-sum two-sided test, corrected for multiple comparisons via FDR). These findings point out the crucial role of nonlinear brain dynamics in order to characterize different brain states; e.g., sleep stages N1–N3 and wakefulness, as the classification performed on the intrinsic manifold significantly outperforms the one performed on the linear dimensionality reduction such as PCA. Furthermore, the successful classification of different brain states on the intrinsic manifold also suggests that it reveals the low-dimensional structure underlying the brain dynamics characteristic to these different brain states.

## Discussion

In this work, we presented a method to reveal the intrinsic manifold underlying brain dynamics measured by fMRI. Patterns of cortical activity in fMRI data exhibit smooth variations over time and hence lie on a low dimensional manifold embedded into the high dimensional space-time representation. In this high-dimensional space-time representation, the dimensionality of the

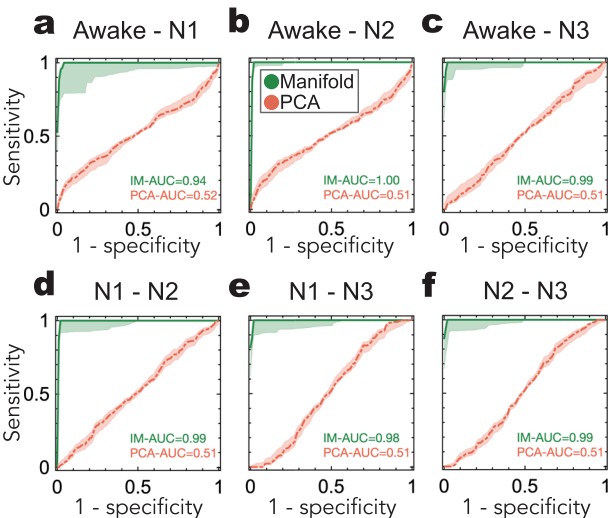

**Fig. 4 Receiver operating characteristic (ROC) for all pairwise comparisons between brain states on the intrinsic manifold of brain dynamics and compared to PCA.** ROC reveals the relationship between sensitivity and 1-specificity. The area under the curve (AUC) indicates how accurately the two compared states can be classified using only one dimension of the embedding space. ROC curves of the classifications performed on the intrinsic manifold (shown in green) and on PCA (shown in red) are illustrated for all pairwise stage comparisons: **a** between wakefulness and N1, **b** between wakefulness and N2, **c** between wakefulness and N3, **d** between N1 and N2, **e** between N1 and N3, **f** between N2 and N3. For all stage pairwise comparisons between wakefulness, N1, N2, and N3, intrinsic manifold yield significantly higher AUC (for all comparisons, *p*-value < 0.001, Wilcoxon Rank-sum two-sided test, corrected for multiple comparisons via FDR) in the first three-manifold dimensions. Shaded areas indicate the distribution of AUC values across different participants, for performed on their intrinsic manifold, whereas red dots correspond to the AUC values in the PCA space. The solid lines indicate the mean of the distribution across classifications and shaded areas indicate their standard error of the mean (N = 18).

data is artificially defined by the number of voxels in the fMRI volume. In order to extract the intrinsic manifold underlying the brain dynamics, our method capitalizes on the description of these dynamics using phase coherence and on the estimation of the underlying low-dimensional manifold using the Laplacian eigenmaps. Our findings reveal several key insights into the nature of human brain dynamics and how the characteristic of these dynamics change in wakefulness and sleep:

Firstly, in this work we reveal the hidden topology underlying brain dynamics and demonstrate that this topology allows for the decoding and the characterization of different brain states such as wakefulness and different NREM sleep stages. The robust decoding of these different brain states on the intrinsic manifold strongly suggests that the brain dynamics governing these brain states exhibit significantly different characteristics of brain activity and these differences can be revealed if the data is mapped to its intrinsic, low dimensional manifold structure. Crucially, our leave-one-subject-out group analysis reveals that the intrinsic manifolds underlying brain dynamics of different participants share a common topology that is primarily constrained by the brain state (wakefulness and different sleep stages) rather than the individual differences between participants. A useful analogy would be to think of the geometry versus the topology of different trees. Every tree has a unique geometry in their structure, yet all trees in a family of trees (i.e., oak trees) together may share a common topology, which is the topology of

an oak tree. Similarly, in our results, no two individual manifolds are the same, yet they share some structural features that constitute their topology, the topology of the intrinsic manifold. The fact that we can significantly accurately classify the sleep stages of one subject based on the training in other subjects' data, strongly implies that there exists a common topology across the different subject intrinsic manifolds, even though the geometry of different subjects' manifolds may be different.

Secondly, our results suggest that the phase synchrony among different brain regions (90 AAL regions in this work) encode crucial information about brain dynamics in different states such as wakefulness and sleep stages, and forms a feature space, which shares a common topology across participants, i.e., an intrinsic manifold.

Finally, a simple classification method (linear SVM) when performed on the intrinsic manifold of brain dynamics leads to an accurate decoding of different brain states (in this study wakefulness and sleep states defined by polysomnography). Decoding on the intrinsic manifold qualitatively improved our own previous decoding attempts, based on SVM with nonlinear (RBF) kernel functions and substantial parameter tuning[24,25], and non-supervised learning methods applied to the same dataset[47] (see Supplementary Tables 2 and 4). This improvement is of particular interest in the sense that our method uses instantaneous data, rather than windows of time points. A more conceptual difference is the underlying assumption on the domain structure of the brain activity data (as estimated by BOLD intensity). Previous efforts used a nonlinear SVM classifier, which seeks to find the best linear separation between classes in an infinite-dimensional feature space. In our approach, we find this linear separation in a low-dimensional embedding of the data ($d = 7$). Considering the nonlinear nature of the low dimensional embedding utilized in our work, our results clearly demonstrate that different brain states occurring in wakefulness and NREM sleep are characterized by nonlinear transformations of brain dynamics.

Like previous efforts[25,26], our approach revealed that the sleep stage most difficult to decode from brain activity was N1. Our classification analysis showed that N1 was mostly confused with wakefulness, a result that goes in line with the current consensus that N1 stage does not represent a clear sleep stage, but it rather consists on vaguely defined mix of wakefulness and sleep[48]. Given that N1 is not consistently defined in EEG polysomnography, it is easy to appreciate the difficulty in characterizing this stage using the fMRI data[22]. The wakeful conscious state is characterized by the dynamical exploration of a rich and flexible repertoire of brain states[49]. This variability of the awake brain dynamics and the unclear boundaries of the definition of N1 play a crucial role in the relatively lower performance of the awake state and N1 classification in comparison to deeper sleep stages. We attribute the superior classification accuracy achieved by our method to our method's ability to capture the nonlinear relationships within the data. When the nonlinearities play a crucial role, the characteristics of different sleep stages are revealed. A quantitative comparison to PCA, the most well-known linear dimensionality method, confirms this conclusion by yielding significantly higher decoding accuracy on the intrinsic manifold (average accuracy of 96%) compared to PCA (average accuracy of 78%). Hence, our findings strongly suggest that the linearity assumptions of brain dynamics, such as those imposed by PCA[10], are suboptimal to reveal the low-dimensional structure underlying the complex brain dynamics captured by the high dimensional functional neuroimaging data such as fMRI.

Previous studies have attempted to reduce the dimensionality of fMRI data by using linear methods such as PCA[10,50] and ICA[24–26]. The objective of these studies was to explore

simultaneous EEG-fMRI recordings in order to devise machine-learning-based algorithms, allowing sleep staging from the fMRI data itself[24–26]. Being aware of the suboptimality of linear methods, the authors took into account nonlinearities in their classifiers (SVM with nonlinear kernels), but not in the dimensionality reduction methods (ICA). Conversely, in this work, we aimed at studying the nonlinear nature of brain dynamics. Indeed, we have shown that, as long as nonlinear aspects of the data are taken into account, wakefulness and individual NREM sleep stages can be classified precisely in a low-dimensional manifold underlying the fMRI signal, achieving a classification accuracy of 96%.

Precise dimensionality reduction of high-dimensional recordings of brain activity can reveal important aspects of the mechanisms and principles employed by the brain[10,51]. Following this rationale, we unravel a common topology of brain dynamics, shared by different participants, that was originally hidden in the high dimensional structure of the data. Our findings show that taking into account the nonlinearities in the data was necessary to capture changes in vigilance, which is in line with a developing appreciation of the importance of nonlinearities in data analyses of sleep, in EEG[52] and fMRI[53].

Taken together, these results not only demonstrate that the intrinsic manifold of brain activity, which in this work estimated from the fMRI BOLD signal, provides a powerful representation of the spatiotemporal dynamics of brain activity, but it also reveals a characteristic feature of brain dynamics that is shared across different participants and is only dependent on different brain states, such as different sleep stages and wakefulness. These findings open the door to investigate characteristic features of various brain states such as drug-induced altered states of consciousness as well as different psychiatric conditions using the intrinsic manifold of brain dynamics.

## Methods

**EEG-fMRI acquisition**. EEG data was recorded via a cap (modified BrainCapMR, Easycap, Herrsching, Germany) continuously during fMRI acquisition (1505 volumes of T2*-weighted echo-planar images, TR/TE = 2080 ms/30 ms, matrix 64 × 64, voxel size 3 × 3 × 2 mm³, distance factor 50%; FOV 192 mm²) at 3 T (Siemens Trio, Erlangen, Germany) with an optimized polysomnographic setting (chin and tibial EMG, ECG, EOG recorded bipolarly [sampling rate 5 kHz, low pass filter 1 kHz], 30 EEG channels recorded with FCz as the reference [sampling rate 5 kHz, low-pass filter 250 Hz], and pulse oximetry, respiration recorded via sensors from the Trio [sampling rate 50 Hz]) and MR scanner compatible devices (BrainAmp MR+, BrainAmp ExG; Brain Products, Gilching, Germany) facilitating sleep scoring during fMRI acquisition[18,19,54].

MRI and pulse artifact correction was performed based on the average artifact subtraction (AAS) method[55] as implemented in Vision Analyzer2 (Brain Products, Germany) followed by objective (CBC parameters, Vision Analyzer) ICA-based rejection of residual artifact-laden components after AAS resulting in EEG with a sampling rate of 250 Hz[19]. Sleep stages were scored manually by an expert according to the AASM criteria[18].

**Participants**. A total of 18 participants were selected (based on the presence of all non-REM sleep stages in their sleep cycle) from a larger dataset in which we measured simultaneously EEG and fMRI, with written informed consent and approval by the local ethics committee (Ethics Committee of the Goethe University of Frankfurt am Main, Germany). Any of the subjects exhibited REM sleep as verified by sleep scoring of the scanner EEG. Participants laid in the scanner during an average time length of 52 min, after being asked not to fight sleep. The subset chosen here was selected based on the condition of spending a minimum of 3 min (87 TRs) in each of the three non-REM sleep stages considered here (i.e., N1–N3), as well as achieving successful EEG, fMRI, and physiological data recording and quality. Inclusion conditions were set to get the maximal representative sampling of each stage, while still maintaining a large number of participants. All participants were scanned during the evening and instructed to close their eyes and lie still and relaxed.

**fMRI pre-processing**. Using statistical parametric mapping (SPM8, www.fil.ion.ucl.ac.uk/spm) echo-planar imaging (EPI) data were realigned, normalized (MNI space), and spatially smoothed (Gaussian kernel, 8 mm³ full widths at half-maximum). Data were re-sampled to 4 × 4 × 4 mm resolution to facilitate the removal of noise and motion regressors.

Cardiac, respiratory (both estimated using the RETROICOR method[56]) and motion-induced noise (inter-scan X-, Y-, Z-displacement and pitch, roll, yaw parameters) were regressed out by least squares. In total, we have modeled 24 motion regressors (6 roto-translations plus their derivatives up to order three). Data were band-pass filtered in the range 0.04–0.07 Hz[42] using a sixth-order Butterworth filter. We did not perform global signal regression and did not apply image censoring methods. Data sets with jumps in the motion regressors (head jerks >0.2 mm inter-scan displacement) were excluded from the cohort analyzed[57].

**Coherence connectivity dynamics**. The functional connectivity dynamics FCD($t_i$, $t_j$) matrix captures the similarities between the functional connectivity matrices FC($t_i$) and FC($t_j$) for each pair of time points $t_i$ and $t_j$. Here we used CCD($t_i$, $t_j$), a frequency-specific adaptation of the FCD matrix that characterizes the time-dependency homogeneity of the synchronization (i.e. coherence) across all brain areas[35]. First of all, we computed the instantaneous phase of the time series by means of the Hilbert transform (*Hilbert.m* in MATLAB_2020b). Then we computed the coherence vector $V(t)$, containing the cosine of the phase difference between all areas of the brain. At each time point, $V(t)$ defines the temporal brain coherence state. The CCD($t_i$, $t_j$) matrix is constructed as the cosine similarity between the coherence vectors at different times:

$$\text{CCD}\left(t_i, t_j\right) = \frac{V(t_i)V\left(t_j\right)}{\left|V(t_i)\right|\left|V\left(t_j\right)\right|} \tag{1}$$

Solid blocks around the CCD matrix diagonal represent epochs of stable coherence states. We then used the CCD matrix, which provides information about the time-dependency of spatial phase coupling dynamics, to construct the time manifolds.

**The RMST**. In order to link states across time, the CCD are pruned into a graph. To do so, first, we need to transform this similarity matrix into a distance matrix as $d(t_i, t_j) = 1 - \text{CCD}(t_i, t_j)$.

The RMST can be used as a way to prune a distance matrix and preserve only local information while being robust to inhomogeneity of sampling, a common problem affecting other global sparsification techniques such as the $k$-nearest-neighbors or the epsilon-ball techniques used in the original publication of the Laplacian eigenmaps method[36]. It involves relaxation of the minimum spanning tree (MST), a problem consisting of finding, within an undirected weighted graph, the subgraph, in which all pairs of nodes are connected by exactly one path (i.e., a tree), which also minimizes the total sum of edge weights. Note that the MST yields a graph with one connected component while minimizing the number of connections of each node. The RMST has been proposed to construct a more informative model of the continuity of the data[44] by relaxing the constraint on the number of connections per node in the MST; i.e. adding more connections at each node if the following condition is fulfilled:

$$mw_{ij} + \gamma\left(d_i^k + d_j^k\right) > d_{ij}, \tag{2}$$

where $mw_{ij}$ is the maximum weight in the shortest path of the initial MST graph between nodes $i$ and $j$, $d_i^k$ is the value of the distance matrix corresponding to the node $i$ and its $k$ nearest neighbor, and $d_{ij}$ is the distance between nodes $i$ and $j$. This condition is applied to each pair of nodes. In order to simplify the solution search, we limited the maximum number of neighbours of each node to $k_n$ ($k_n = 5$ in this study) while fixing the distance $d_i^1$ of each node to its nearest neighbor. The $\gamma$ is the relaxing parameter and allows to add of weaker connetions ($\gamma = 3$ in this study).

**Dimensionality reduction and Laplacian eigenmaps**. Traditionally, the most commonly known dimensionality reduction approaches are linear methods, such as PCA, factor analysis and classical scaling. However, these approximations fail when applied to complex nonlinear datasets[58]. Broadly, manifold learning techniques rely on the assumption that the high-dimensional data points lie on a lower-dimensional manifold and aim to estimate this low-dimensional manifold underlying the high-dimensional data points. Let X = $\{\mathbf{x}_1, \mathbf{x}_n\} \in \mathcal{M} \subset \mathbb{R}^D$ be the high-dimensional observations. Manifold learning algorithms estimate a function $\mathbf{y}_i = \boldsymbol{f}(\mathbf{x}_i)$ such that $\forall \mathbf{x}_i \in \mathcal{M}, \boldsymbol{f} : \mathcal{M} \mathbb{R}^d$ with $d \ll D$. This estimated function maps each data point from the manifold in the high-dimensional space $\mathbb{R}^D$ to the embedding in a lower-dimensional space $\mathbb{R}^d$, with dimension $d \ll D$. This low-dimensional representation has coordinates $\{\mathbf{y}_1, \mathbf{y}_n\} \in \mathcal{Y}$. Different manifold learning algorithms differ in their estimation of the mapping function $\boldsymbol{f}$. However, they all share a basic pipeline structure consisting in: (1) computing a measure of the relationship between each pair of data points $S(\mathbf{x}_i, \mathbf{x}_j)$, (2) defining a similarity matrix $\mathbf{H}$ based on these pairwise relationships, and (3) estimating the eigenvalues $\{\lambda_1, ..., \lambda_d\}$ and eigenvectors $\{\mathbf{v}_1, ..., \mathbf{v}_d\}$ of $\mathbf{H}$. The $d$-dimensional representation $y$ in $\mathbb{R}^d$ is given by the eigenvectors of $\mathbf{H}$ corresponding to the $d$ smallest, non-trivial eigenvalues.

PCA can be seen as a linear manifold learning technique, in which the matrix similarity $\mathbf{H}$ is defined as the covariance matrix of the data, and its eigenvectors provide the function basis onto which the high-dimensional observations are projected. Nonlinear manifold learning algorithms, in contrast, obtain the intrinsic structure directly using those eigenvectors.

The rationale underlying the LE algorithm is to find a low-dimensional embedding that best preserves the local structure of the data. LE uses the discrete version of the Laplace–Beltrami operator, also known as graph Laplacian (**L**) as the similarity matrix **H**, and estimates its eigenvectors. Here we studied the implementation of LE to a dataset of fMRI BOLD signal of human participants among different wakefulness–sleep stages. Note that due to the nature of the nonlinear dimensionality reduction, the mapping function from the high-dimensional to the low-dimensional space is not available. Therefore, it cannot be reversed to project the data points back into the high-dimensional space. Unlike linear methods like PCA, the nonlinear manifold learning methods reveal directly the low-dimensional coordinates of the data points, and the nonlinear mapping is implicitly applied in the estimation of these coordinates by the method.

The PCA was performed on the pre-processed BOLD time-series, with AAL parcellation, and time-points as observations (*pca.m* in MATLAB_2020b). The LE is part of the intrinsic manifold computation and its implementation is explained in the "Results" subsection "Revealing the intrinsic manifold of brain dynamics".

**Group analysis of individual manifolds**. In order to assess how the topology of the individual intrinsic manifolds generalized across participants, we aligned the low-dimensional embeddings using a linear transformation. For each subject, the linear transformation matrix **T** is defined as follows:

$$T^* = \arg \min_{\substack{T \\ s.t. \\ T^\top T = I}} \|TX_s - X_{\mathrm{ref}}\|_F,$$ (3)

where $X_s$ denotes the matrix with columns $x_i = [\arg \min(\|u_i\|_2), \arg \max(\|u_i\|_2)]$, corresponding to the data-points with minimum and maximum $\ell_2$—the norm for each sleep stage $i$ and subject $s$. Finding the best transformation $T^*$ corresponds to solving for best rotation, translation and scale between the two sets of points. Here we used procrusted analysis[59] with its MATLAB_2020b implementation *procustes.m*.

**SVM classifier**. In order to evaluate how well the intrinsic manifolds represent the structure underlying brain activity during sleep and wakefulness, we tested whether different stages of the wakefulness–sleep cycle can be accurately classified on the manifold underlying brain activity. To this end, we utilized a SVM classifier. SVMs are a set of supervised learning tools commonly used in various classification problems[60]. In this work, we used a simple linear SVM, so that nonlinearities were only accounted by the low-dimensional intrinsic manifold.

In order to estimate the efficiency of the intrinsic manifolds for classification of different sleeps stages, and how well this classification generalizes to independent datasets, we used cross-validation: we split the data into different subsets which were used exclusively either in the training or testing phase of the cross-validation. In the case of individual subject manifolds, we applied 10-fold cross-validation for each stage pairwise comparison, dividing the data in each stage into 10 parts, using 1 of those parts as the test set and the 9 remaining parts as the training set. The resulting accuracy was averaged across the 6 possible combinations to obtain a distribution across the 18 participants. For the group manifold computed by combining the data of all participants, we performed the classification using the leave-one-subject-out cross-validation, training each time with 17 participants and testing with the remaining one, and obtaining the distribution of accuracies across all the 18 classifications performed.

In the main text, the accuracies of classification are reported sometimes as the average across different stage-to-stage comparisons. In those cases, the uncertainty is calculated as follows (in the case of the awake stage):

$$e_{\mathrm{Awake}} = \frac{1}{3} \sqrt{e_{A-N1}^2 + e_{A-N2}^2 + e_{A-N3}^2},$$ (4)

where $e_{A-X}^2$ is the average classification accuracy uncertainty for each awake stage comparison (standard deviation across subjects of the mean cross-validation accuracy).

We also tested the manifold decoding capacity, using multi-class classification on single time points. We trained four one-vs-all classifiers, in which data samples were binarily categorized as belonging to one stage or not. During the training phase, a posterior distribution of the scores was fitted for each classifier. During the testing phase, the posterior distribution was used to determine the probability of each test sample belonging to each sleep stage. All samples were fed to the four one-vs-all classifiers, using 10-fold cross-validation in the case of individual manifolds, and leave-one-subject for the group manifold, and the class with the highest probability was used to appoint the predicted class.

**Statistical significance analysis**. In order to test for statistical significance, we used nonparametric statistical tests. Note that due to the fact that pre-processing includes low-pass filtering to compute the narrowband instantaneous phase, we might be introducing a partial, low-frequency structure to the data. As suggested in ref. [61], phase randomized surrogates of the time series is used to obtain a *p*-value against the null hypothesis that clustering of data points in the intrinsic manifold space is due to the low-pass filtering and not to real similarities.

To this end, we performed a Monte Carlo permutation test by comparing the results of classification of the original data to the classification on 1000 random-sampled permutations of the fMRI BOLD time series. The accuracies of the phase

randomized surrogates were used to generate a distribution from which the *p*-value is computed as

$$p = \frac{\sum_{i}^{N_{\mathrm{perm}}} f\left(a, a_{\mathrm{perm}}^i\right) + 1}{N_{\mathrm{perm}} + 1},$$ (5)

where $N_{\mathrm{perm}}$ is the number of permutations of the Monte Carlo test, $a$ is the accuracy obtained by the SVM in the original data, and $a_{\mathrm{perm}}^i$ is the accuracy obtained by the SVM in the $i$th Monte Carlo simulation, and

$$f\left(a, a_{\mathrm{perm}}^i\right) = \begin{cases} 1, & a < a_{\mathrm{perm}}^i, \\ 0, & a \geq a_{\mathrm{perm}}^i. \end{cases}$$ (6)

The phase randomization was performed by first obtaining the Fourier spectrum of the time series, adding uniformly distributed shifts to the phase. The negative frequencies' phases are kept equal to their positive counterpart to have a real surrogate signal, and the DC component is preserved intact. The shift value is constant across ROIs. The inverse Fourier transform of the phase-randomized spectrum gives the phase-randomized surrogate.

**Receiver-operating characteristic**. In binary classification problems, amongst other measures of performance, we obtained the true positive rate (number of correctly classified cases of one of two classes, divided by the total number of classified items) and false positive rate (number of wrongly classified cases of the other class, divided by the total number of classified items). The ROC curve is the result of plotting these two measures for all possible discrete thresholds in the dimension over which we are classifying the data. If two classes are completely separated in one given dimension, then the best possible prediction would give a true positive rate of 1 and a false positive rate of 0. Nonetheless, in problems dealing with noisy observations, two classes are not perfectly separable for any threshold value. For this reason, we exhaustively measured the rate of true positives and false positives along the dimension of interest and compute the ROC curve by varying a classification decision threshold. The area under this curve (AUC) is an indicator of how separable those two distributions are and consists of the integral of the ROC curve. An AUC value of 1 indicates that the two classes are perfectly separable, whereas an AUC value of 0.5 indicates that the two classes have completely overlapping distributions.

**Reporting summary**. Further information on research design is available in the Nature Research Reporting Summary linked to this article.

## Data availability
All data generated in this study are available from the corresponding author upon reasonable request.

## Code availability
All custom scripts used in this study were written in MATLAB_2020b, and are available in https://github.com/joanrue/intrinsic-manifolds[62].

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

## Acknowledgements

J.R.-Q. is funded by the Fundació Catalunya—La Pedrera Masters of Excellence Fellowship. M.L.K. and S.A. are supported by the ERC Consolidator Grant CAREGIVING (no. 615539) and Center for Music in the Brain, funded by the Danish National Research Foundation (DNRF117). G.D. is supported by a Spanish national research project (ref. PID2019-105772GB-I00 /AEI10.13039/501100011033 MCIU AEI) funded by the Spanish Ministry of Science, Innovation and Universities (MCIU), State Research Agency (AEI).

## Author contributions

J.R.Q., S.A., M.L.K., and G.D. designed the methodology and the analysis. E.T. and H.L. collected and preprocessed the data. A.S. aided in the interpretation of the results. S.A., J. R.-Q., A.S. and M.L.K. wrote the manuscript. All authors reviewed the manuscript.

## Competing interests

The authors declare no competing interest. E.T. is an Editorial Board Member for *Communications Biology*, but was not involved in the editorial review of, nor the decision to publish this article.
