## [Peer Review File · Communications Biology]

Reviewers' Comments:

Reviewer #1:

Remarks to the Author:

This is a rare occasion on which, as a reviewer, I can say I really like the paper very much, almost in every aspect. The overall objectives of the paper are exciting and important, the results are in line with those objectives, and the results sustain the discussion. I have no objections to the publication of this paper. I highly congratulate the authors for this magnificent work.

I only include a series of suggestions to clarify specific points or perhaps enhancing the interpretation of some of the presented results:

1) The authors mention previous efforts to classify sleep stages using fMRI data (e.g. Tagliazucchi Neuroimage 2012).

However, the paper is missing a direct comparison and discussion of the results obtained in that paper versus the ones obtained here. This is furthermore important if we consider that both papers use the same dataset.

2) In the same vein, I found interesting (and intriguing!) that the method presented in this paper has the lowest performance when the awake state is compared with the different sleep stages. These results are contrary to the ones shown in Tagliazucchi 2012, and very counterintuitive if we take into consideration the electrophysiological similarities of the different stages.

3) The authors report a 'Common intrinsic manifold topology' across participants with a group classification performance of 76%. However, by visually inspecting the individual embeddings is hard to imagine a generalization across participants. Each of the subspaces corresponding to a sleep/awake stages are easy to separate on an individual basis, but they don't seem to occupy an analogous subspace across participants.

In that sense, I wonder if the group results emerge from a bimodal distribution (that is, subjects that either entirely resembles the group and others that for which the subspaces corresponding to each stage lie in a completely different location) or a homogenous similarity. If the former is true, it would be worth investigating if these individual subspaces can somehow be interpreted as an 'individual fingerprint' that can be used to identify subjects univocally. Also, it would be worth discussing if some mapping can be applied across subjects to validate the real universal value of the subspaces corresponding to each sleep/awake stage.

Reviewer #2:

Remarks to the Author:

This paper applies a nonlinear dimensionality reduction method (Laplacian eigenmaps) to fMRI data in sleep and wake (specifically to phase coherence matrices). The main results are that wake and non-REM sleep states occupy distinct areas in the reduced dimensional space, and that these states can be reliably distinguished in that space. Since wake and sleep states can be distinguished at the group level, it is argued that the reduced representation captures some "invariant" and "intrinsic" aspects of human brain dynamics. The topic is one of wide interest, but I have various concerns about the results and the methods, so some of the conclusions appear overblown. Some issues are probably just matters of presentation, others perhaps not. The following points should be addressed:

1. It is claimed that "the intrinsic manifolds of all participants share a common topology", but what

IS that topology? From Fig. 2A-H (and more so all subjects in Fig. S1) there is clearly *substantial* variability across subjects. It seems very difficult indeed to claim that there is a clear common topology here. Fig. 2I and Fig. S3 show an embedding combining all subjects together, which bears little resemblance to the subject-level data. The main common feature appears to be a highly radial organization of the points, though I'm not sure what that means, and the radial orientations differ widely between subjects. This needs to be clarified.

2. So what is the proposed dimensionality of the intrinsic dynamics? The choice of $d=3$ is used for the classifications, though there's also $d=10$ in the supplementary figures, and a scan over dimensions in Figures 3F and 3G. Perhaps a case can be made for $d=3$ being near the knee in Fig. 3F (though probably a poorer choice than $\sim 5-8$), but there is no evidence that any 'optimal' dimensionality is found at the group level (Fig. 3G), since the accuracy seems far from plateauing. With 18 subjects and no apparent optimal dimensionality for a 25 dimension embedding, is the method failing to converge on any useful commonality?

3. It is strange for the classification problem to be tackled as a set of pairwise SVMs for states A vs B for all pairs, rather than state A vs not-A (for all states), or some other non-binary classification that handles all the states together. The headline claim that the method can "decode" the states with an average accuracy of 93% seems inadequately substantiated -- it is standard for such problems to present the confusion matrix showing the probabilities of correct classification and incorrect classification for all possible combinations.

4. Is Figure 1G a schematic or an actual example of a 2D embedding? Why are the points arranged on a grid when they are not obviously gridded in the actual results? If the brain takes the path Awake  S1  S2  S3, does that mean it starts in the bottom left corner, heads up to the top left, then backtracks halfway and heads to the right? Why essentially no points 'off axis'?

5. After presenting Fig. 2 briefly, the results jump into decoding of states. But there's a lot going on in Fig. 2 that should be explained in more detail. The paper does not address whether the data exhibits sensible trajectories through the reduced space. Is each point one snapshot in time? In what temporal order are they linked? Does the brain traverse in and out of the origin along different radial paths for each state (or around loops for e.g. Fig. 2E), or does it take 'shortcuts' from one arm to the next? Do these trajectories make sense in light of all the existing work on transition probabilities for sleep states etc.? Also the PCA results should at least be mentioned before moving on.

6. p11, "These findings point out the crucial role of nonlinear brain dynamics in order to characterize different brain states" -- this seems to refer to one sense of nonlinearity (of some abstract embedding), but the usual way to test whether brain dynamics are nonlinear is to compare to linear surrogate data. e.g., would linear surrogates on the original time series, if fed through the same analysis pipeline, end up with the same result? There are numerous methods for performing such analyses (e.g. Breakspear et al. 2004 HBM, Liegeois et al. 2017 NeuroImage). I note the authors did use permutation testing that shuffled the time series, but this is known to be a strawman surrogate for dynamic functional connectivity, cf. Liegeois et al. 2017.

7. The precise workflow from the coherence to the reduced space is a little unclear. So each point in time has a phase coherency matrix (Fig. 1F), and the coherence connectivity dynamics matrix (not shown) captures the similarities between all the phase coherency matrices at all pairs of time points. Then this is pruned into a graph thus linking states across time. There's a mention of a distance matrix in the RMST section, but what is the distance metric here? How much of the smoothness of the manifolds is due to imposing some continuity over time? The way that time is encoded in the CCD matrix as all the pairwise comparisons makes it hard for me to intuitively grasp how that summary matrix can be decomposed into distinct states across time, vs the schematic Fig. 1F-G where I can think of each phase coherency matrix (==time point) occupying a single point in the reduced space.

8. How exactly was the PCA calculated? What was the input?

9. Did the data include REM sleep? If not, why not? If yes, how were those periods handled?

10. The Methods section "Intrinsic manifolds of brain activity" should be in the Results to better guide the reader through what was done (it does a better job of summarizing the workflow than the description of Fig. 1 in the Results).
11. Figures and supplementary figures are presented together without saying what each shows distinctly (reader needs to work out what is different in the paired supplementary figure).
12. The Atasoy et al. connectome harmonics work is cited as an application of Laplacian eigenmaps, but that was eigenfunctions of the graph Laplacian (those papers never referred to 'eigenmaps'); related to but not the same as the dimensionality reduction technique here. If this is relevant background, then also relevant would be Tokariev et al. 2019 Nature Commun relating cortical eigenmodes to sleep states.
13. "the intrinsic manifold yields a smooth, continuous representation of the brain dynamics" -- well it's clearly not continuous in the strict sense since there are discrete points, and hence cannot be smooth either.
14. Fig. 3, there are two panel Es, and in the caption "D) and H)" is meant to be "G) and H)" (after fixing the E duplication).
15. p8, accuracy of "68±1%" seems peculiarly precise, how is the uncertainty calculated?
16. "PCA was unable to capture the structure underlying brain dynamics" -- well, it was weakly but significantly different from chance so it partly captures something (PCA clearly performed poorly, but don't overcook it).
17. Does PCA's failure come down to nonlinearity or something else? Just because a NL method does well and a linear method does not, doesn't mean that the fault lies with the linearity.
18. It is mentioned that this classifier outperforms previous efforts to detect sleep states from fMRI, but by how much? This should be quantified.
19. Regarding dynamics existing on subspaces of the high-dimensional phase space, this idea of course goes back a long way, at the very least to Haken, and obviously all the work on modeling neural masses and fields etc. And even ignoring such theoretical efforts, fMRI obviously has spatial smoothness so no one should think of every voxel being independent.
20. Abstract mentions "nonlinear transitions", but "transitions" doesn't appear again. In what sense are the transitions nonlinear? Also not clear that transitions are even studied here.
21. The Methods presents the Hilbert transform to a peculiar depth; it could be shortened. Did the authors just use MATLAB's `hilb()`? If not, why not?
22. "We first extracted the instantaneous phase signal of the ultraslow fluctuations (0.04 - 0.07 Hz)" vs "Data were band-pass filtered in the range 0.01–0.1 Hz" -- which was it? And why 0.04-0.07 Hz anyway?
23. p18, typo "RMTS"
24. Why does the dimensionality reduction section review other methods? If including that material, it should be earlier in the paper to provide more context, or else just remove it to keep things simpler.

Reviewer #3:

Remarks to the Author:

The current study presents a novel method for calculating low dimensional manifolds in fMRI data

and applies it to data at different sleep/wake cycles. I thought the paper was interesting considering recent trends to investigate low dimensional manifolds in fMRI data. In general the paper was well written with clear aims etc., I have a few comments below.

Was any further fMRI preprocessing performed? The methods seem very sparse compared to common practice where nuisance variables such as head motion are regressed (Ciric et al., 2017).

Projecting the dimensions back into brain space would be helpful for the reader to contextualize the manifolds - which are by nature quite abstract - e.g., what the authors have done in Figure 1E.

The paper has a few typos, e.g., page 21 paragraphs regarding SVM, 'splitting' should probably be 'split', 'resting' should probably be 'remaining'.

I am wondering why the predictions were on a stage-by-stage basis and not all sleep categories in a single classification model. Wouldn't an 'all in one' analysis demonstrate the point in a simpler fashion? (which could have the current analyses as follow-ups)

Relatedly, is it possible that different features contributed to the classification accuracy depending on the given contrast? Is it possible to, for example, plot the SVM weights to see which features contributed?

The authors compare the current method to PCA - another method used in fMRI to identify low dimensional states. Would a more appropriate 'null' model be data that is high dimensional?

Dear Reviewers,

We wish to thank you for your constructive comments and suggestions. We have considered all your points during revision and the following is a point-by-point explanation of all the changes made. After addressing these raised questions, we believe that our revised manuscript has been improved in terms of its clarity, readability, and ultimately in its contribution to the neuroscience community.

In addition to this response letter, we include two versions of the revised manuscript: a "marked" version that highlights all of the changes made since the original submission and a "clean" version that includes all changes but without any highlights. In the "marked" version, all newly added parts are marked as **blue**, whereas the edited parts are marked as **green**, and the corresponding Reviewers' comment is marked as comment throughout the manuscript next to each highlighted change.

Following your suggestions, we have made two main modifications in our revised manuscript, which we believe have substantially improved the quality of our study and our manuscript. Firstly, we revised our method for computing the intrinsic manifold for the group data. In the revised manuscript we compute the group manifold by registering individual subject manifolds and not by estimating a single manifold using all subjects' data altogether. This change in the estimation of the group manifold by mapping individual manifolds to the same space has significantly improved the accuracy of classification at the group level. Secondly, we extended our classification experiments to also include multi-class evaluation. Following your suggestions, we demonstrate the accuracy and robustness of the intrinsic manifold approach for classification by also performing all-vs-all multi-class decoding of different sleep stages and wakefulness based on multi-class support vector machines (SVM).

Please do not hesitate to contact us if we can provide any further clarification.

Sincerely,

Joan Rué-Queralt and Selen Atasoy (on behalf of all co-authors)

Detailed Response for Reviewers on the Manuscript

COMMSBIO-20-1817-T

RESPONSE TO REVIEWER 1

Comment 1.0: *This is a rare occasion on which, as a reviewer, I can say I really like the paper very much, almost in every aspect. The overall objectives of the paper are exciting and important, the results are in line with those objectives, and the results sustain the discussion. I have no objections to the publication of this paper. I highly congratulate the authors for this magnificent work. I only include a series of suggestions to clarify specific points or perhaps enhancing the interpretation of some of the presented results:*

Answer 1.0: We thank the Reviewer very much for this positive remark and appreciate their support. We believe that through their and other Reviewers' suggestions, the presentation and quality of our paper has further improved in the revised manuscript.

Comment 1.1: *The authors mention previous efforts to classify sleep stages using fMRI data (e.g. Tagliazucchi Neuroimage 2012). However, the paper is missing a direct comparison and discussion of the results obtained in that paper versus the ones obtained here. This is furthermore important if we consider that both papers use the same dataset.*

Answer 1.1: We thank the Reviewer for pointing out this opportunity to improve the clarity of our presentation. We agree that our comparison to the cited previous efforts has not been discussed in sufficient detail. It is important to note there are some practical and conceptual differences between previous efforts and the current approach, which make a direct comparison of the classification accuracies of the two approaches rather difficult. These are:

The first practical difference is that, using the same dataset, the authors in *Tagliazucchi et al. Neuroimage 2012* classify windows of timepoints (1-4 minutes long). Unlike this window-based approach, our method classifies each individual timepoint in the fMRI data. Thus the classification accuracies in that publication refer to a window of time points, whereas in our work they are estimated for each individual time point. Another important practical difference is that in their work, Tagliazucchi and colleagues used 5-fold cross-validation, and reported the *maximum* accuracy among the 5 test sets. In our group analysis, we take a more conservative approach and perform leave-one-subject-out cross-validation. This allows us to make statements on the generalizability of the learned classifiers among unseen subjects. Importantly, we report the *mean* accuracy over test sets and not the maximum. Yet, despite the conservative classification approach taken by our method, we followed the

Reviewer’s suggestion and included a table for comparison of the two methods’ classification accuracies. Below we report the results of *Tagliazucchi et al. Neuroimage 2012*, in comparison to our results on group manifolds, and on individual manifolds. We note that a fair comparison between the methods in terms of their classification accuracies is only possible when the accuracy of the intrinsic manifold method using individual manifolds is compared to the results presented in *Tagliazucchi et al. Neuroimage 2012*, as only in that comparison, data from the same subjects are included in both training and testing sets for both methods.

	Tagliazucchi 2012 (TW: 1 min; max accuracy out of 6-fold – see their TABLE 5)	Intrinsic manifold (GROUP, single time-points; mean accuracy out of leave-one-subject-out)	Intrinsic manifold (INDIVIDUAL, single time-points; mean accuracy out of 6-fold cross-validation)
Awake – N1	0.76	0.82	0.96
Awake – N2	0.88	0.91	0.99
Awake – N3	0.89	0.96	0.99
N1 – N2	0.83	0.90	0.99
N1 – N3	0.93	0.97	0.99
N2 – N3	0.87	0.94	0.99

Finally, a more conceptual difference worth noting is the underlying assumption on the domain structure of the brain activity data (as estimated by BOLD intensity). *Tagliazucchi et al. Neuroimage 2012* use a nonlinear SVM Classifier, which seeks to find the best linear separation between classes in an infinite dimensional feature space. In our approach, we find this linear separation in a low dimensional embedding of the data (d=7).

Following the Reviewer’s suggestion, we have now included a detailed explanation of the comparison with previous efforts including all the above-mentioned points, in the revised manuscript on page 16. We have also included the above table in the supplementary material, see Table S4.

Comment 1.2: *In the same vein, I found interesting (and intriguing!) that the method presented in this paper has the lowest performance when the awake state is compared with the different sleep stages. These results are contrary to the ones shown in Tagliazucchi 2012, and very counterintuitive if we take into consideration the electrophysiological similarities of the different stages.*

Answer 1.2: We thank the Reviewer for pointing out this important observation. As now clarified in the revised version of the manuscript, like the previous efforts (*Tagliazucchi and Laufs Neuron 2014*

and Altmann *et al. NeuroImage* 2016), our approach yield that the sleep stage N1 as the state hardest to decode from brain activity. Our classification analysis showed that N1 was mostly confused with wakefulness, a result that goes in line with the current consensus that N1 stage does not represent a clear sleep stage, but it rather consists on vaguely defined mix of wakefulness and sleep (Carskadon and Dement, *Principles and practice of sleep medicine Ch.2* 2011). Given that N1 is not consistently defined in EEG polysomnography, it is easy to appreciate the difficulty in characterizing this stage using the fMRI data (Stevner *et al. Nat. Comm.* 2019).

We believe that the relatively lower performance of classification for wakefulness and N1 is due to the large variability of brain states in the awake state and the unclear definition of N1. Wakeful conscious state has been mainly characterized by the dynamical exploration of a rich and flexible repertoire of brain states (Barttfeld *PNAS* 2015). This variability in the dynamics is blurred out in the results in Tagliazucci *et al. Neuroimage* 2012 due to the long time-windows used for averaging before defining the brain states to be decoded. However, in our method using every single time point, this variability of the awake brain dynamics is clearly observable and plays a crucial role in the relatively lower (95%) performance of the presented approach in the awake state classification in comparison to deep sleep stages (98% and 100%, for N2 and N3 respectively). We have extended our interpretation of these results in the discussion section of the revised manuscript (see page 16).

Comment 1.3: *The authors report a ‘Common intrinsic manifold topology’ across participants with a group classification performance of 76%. However, by visually inspecting the individual embeddings is hard to imagine a generalization across participants. Each of the subspaces corresponding to a sleep/awake stages are easy to separate on an individual basis, but they don’t seem to occupy an analogous subspace across participants.*

In that sense, I wonder if the group results emerge from a bimodal distribution (that is, subjects that either entirely resembles the group and others that for which the subspaces corresponding to each stage lie in a completely different location) or a homogenous similarity. If the former is true, it would be worth investigating if these individual subspaces can somehow be interpreted as an ‘individual fingerprint’ that can be used to identify subjects univocally. Also, it would be worth discussing if some mapping can be applied across subjects to validate the real universal value of the subspaces corresponding to each sleep/awake stage.

Answer 1.3: We thank the Reviewer for pointing out this interesting interpretation of our results. We believe that this is a very important research direction and thank the Reviewer for this insight. Our conclusion about the common intrinsic manifold topology is based on the classification accuracy on our leave-one-subject out experiments. It is possible that these results emerge from a bimodal distribution, however exploring these individual differences, and analyzing the differences between

the individual and group manifolds are beyond the scope of this manuscript. However, inspired by the Reviewer's comment, we reformulated our group analysis in the revised manuscript. Given that we are interested in the manifolds' topology, rather than its exact state-space location, we registered all individual intrinsic manifolds to a common space by applying an affine transformation (shown in Figure 2I in the revised manuscript). The group manifold is then given by this registration of all individual manifolds. Remarkably, using this new approach our results not only validate the generalizability of the intrinsic manifolds across subjects as shown by the high group-analysis classification accuracies, but also yield higher classification accuracy than our previous approach. This is further explained on pages 8 and 22.

We thank the Reviewer for the inspiration of the reported improvement in our methods and also refer to our comment to Reviewer 2.1 for further discussion on this subject.

RESPONSE TO REVIEWER 2

Comment 2.0: *This paper applies a nonlinear dimensionality reduction method (Laplacian eigenmaps) to fMRI data in sleep and wake (specifically to phase coherence matrices). The main results are that wake and non-REM sleep states occupy distinct areas in the reduced dimensional space, and that these states can be reliably distinguished in that space. Since wake and sleep states can be distinguished at the group level, it is argued that the reduced representation captures some "invariant" and "intrinsic" aspects of human brain dynamics. The topic is one of wide interest, but I have various concerns about the results and the methods, so some of the conclusions appear overblown. Some issues are probably just matters of presentation, others perhaps not. The following points should be addressed:*

Answer 2.0: We thank the Reviewer for their constructive and thoughtful comments. We have addressed all of the raised concerns of the Reviewer, which in our opinion significantly improved the quality of our manuscript.

Comment 2.1: *It is claimed that "the intrinsic manifolds of all participants share a common topology", but what IS that topology? From Fig. 2A-H (and more so all subjects in Fig. S1) there is clearly *substantial* variability across subjects. It seems very difficult indeed to claim that there is a clear common topology here. Fig. 2I and Fig. S3 show an embedding combining all subjects together, which bears little resemblance to the subject-level data. The main common feature appears to be a highly radial organization of the points, though I'm not sure what that means, and the radial orientations differ widely between subjects. This needs to be clarified.*

Answer 2.1: We thank the Reviewer for pointing out this interesting and important observation about our results. Our claim that the intrinsic manifolds of different subject share a common topology is driven by the fact that our leave-one-subject out experiment yields significantly accurate classification accuracy. This means that using the intrinsic manifold of the sleep data that does not belong to that particular subject as a training set, our method is able to significantly accurately classify each time point of a new subject's fMRI data, which would not be possible if there was not a common topology shared by the intrinsic manifolds across different subjects. However, we agree with the Reviewer that our results and the use of the word 'topology' call for further clarification. To this end, a useful analogy would be to think of the geometry vs topology of different trees. Every tree has a unique geometry in their structure, yet all trees in a family of trees, (i.e., oak trees), together may share a common topology, which is the topology of an oak tree. Similarly, in our results, no two individual manifolds are the same, yet they share some structural features that constitute their topology and make them recognizable by the SVM classifier. The fact that we can significantly accurately classify the sleep stages of one subject based on the training in other subjects, strongly implies that there exists a common topology across the different subject intrinsic manifolds, even though the geometry of different subjects' manifold may be different. Following the Reviewer's suggestions, we added a detailed clarification of this point in the revised version of the manuscript, see page 15.

Please also refer to our comment to Reviewer 1.3.

Comment 2.2: *So what is the proposed dimensionality of the intrinsic dynamics? The choice of $d=3$ is used for the classifications, though there's also $d=10$ in the supplementary figures, and a scan over dimensions in Figures 3F and 3G. Perhaps a case can be made for $d=3$ being near the knee in Fig. 3F (though probably a poorer choice than $\sim 5-8$), but there is no evidence that any 'optimal' dimensionality is found at the group level (Fig. 3G), since the accuracy seems far from plateauing. With 18 subjects and no apparent optimal dimensionality for a 25 dimension embedding, is the method failing to converge on any useful commonality?.*

Answer 2.2: We thank the Reviewer for pointing out this opportunity to improve the clarity of our presentation. In the previous version of the manuscript, the intrinsic manifold parameters were optimized to maximize the 1-vs-1 SVM accuracy of classification. As pointed out by the Reviewer, the results of the classification did not plateaued when increasing the manifold dimensionality. In the revised version of the manuscript, the intrinsic manifold parameters were optimized to maximize the multi-class SVM accuracy of classification, as also suggested by Reviewer 3 in comment 4. When considering the multi-class classification accuracy, our results yield the optimum dimensionality of the intrinsic manifold to be $d=7$. This optimal dimensionality is evidences by the 'knee' shape, and

has been tested for statistical significance ($p < 0.05$, Wilcoxon Rank-sum two-sided test, corrected for multiple comparisons via FDR). This fact has been clarified in the revised version of the manuscript (see pages 8 and 10).

Comment 2.3: *It is strange for the classification problem to be tackled as a set of pairwise SVMs for states A vs B for all pairs, rather than state A vs not-A (for all states), or some other non-binary classification that handles all the states together. The headline claim that the method can "decode" the states with an average accuracy of 93% seems inadequately substantiated -- it is standard for such problems to present the confusion matrix showing the probabilities of correct classification and incorrect classification for all possible combinations..*

Answer 2.3: We thank the Reviewer for pointing out this important point. We agree with the Reviewer that an additional one-vs-all sleep stage type of classification is more commonly used and would be possibly more adequate for any claims done about decoding brain states. Following the Reviewer's suggestion, we have included an additional one-vs-all classification analysis in the revised version of the manuscript, explained in detail in the Methods section (see pages 22 and 23). Remarkably, even in this multi-class classification approach, the intrinsic manifold representation yields an average accuracy of 96%, which is now reported on page 7 in the revised manuscript.

Comment 2.4: *Is Figure 1G a schematic or an actual example of a 2D embedding? Why are the points arranged on a grid when they are not obviously gridded in the actual results? If the brain takes the path Awake  S1  S2  S3, does that mean it starts in the bottom left corner, heads up to the top left, then backtracks halfway and heads to the right? Why essentially no points 'off axis'?*

Answer 2.4: We thank the Reviewer for this important question. Firstly, we would like to clarify the presentation in Figure 1G. This figure was included into the manuscript for the schematic demonstration of the presented approach and illustrates the changes in phase coherency throughout the intrinsic manifold. Figure 1G shows the phase coherency at different points of a 2D embedding for a representative subject. To this end, we have defined 2D bins using the first two manifold dimensions, and computed the average phase coherency of data points in those bins. It is worth noting that in order to have an equal representation for each sleep state, we take the same number of time points for each sleep stage and wakefulness from the dataset, which may lead to some gaps in the temporal continuity of the data. For example, for one subject, we can pick data points belonging to N1 after awake state, and some N1 time-points before the participant is waking up, if the number of continuous time-points just right after awaking is not large enough (in our case 87 TRs, i.e. approximately 3 minutes of scanning time). Thus, the focus of this work is on sleep stage

categorization rather than the temporal transitions between sleep stages. However, we do appreciate the relevance of studying the temporal trajectories, which we have planned to conduct in our future research. In the revised manuscript we have further clarified the purpose of Figure 1G, see page 7. Importantly, inspired by the Reviewer's question, in the revised version of the manuscript we have included a new figure (Figure S4) and a video (Video S5) representing the temporal evolution of the brain state in the manifold. Surprisingly, we discovered that brain activity exhibits both intra-stage smooth transitions and inter-stage sharp jumps in its intrinsic manifold, as clearly illustrated in our supplementary video (Video S5). Specifically, these new results show how temporal traces belonging to one class (sleep stage) are clustered separately in the intrinsic manifold, even though the temporal structure of the data is disrupted, inducing 'shortcuts' between branches of the structure. A detailed explanation on Figure S4 and Video S5 has been included on page 6.

Comment 2.5: *After presenting Fig. 2 briefly, the results jump into decoding of states. But there's a lot going on in Fig. 2 that should be explained in more detail. The paper does not address whether the data exhibits sensible trajectories through the reduced space. Is each point one snapshot in time? In what temporal order are they linked? Does the brain traverse in and out of the origin along different radial paths for each state (or around loops for e.g. Fig. 2E), or does it take 'shortcuts' from one arm to the next? Do these trajectories make sense in light of all the existing work on transition probabilities for sleep states etc.? Also the PCA results should at least be mentioned before moving on.*

Answer 2.5: We appreciate the Reviewer's insightful comment. In this work, we did not explore transitions, as temporal continuity is disrupted during the analysis to favor the presence of equal time points in each sleep stage and wakefulness, as explained in our response to Reviewer's previous comment (R2.4). We have, however, added a Supplementary Figure (Fig S4) in order to explore the temporal path of brain dynamics on the intrinsic manifold. Figure S4 shows the intrinsic manifold labeled according to the temporal evolution. Remarkably, these newly added results reveal that there exists temporal continuity within one stage, such as N1, or N2, or wakefulness, yet different stages are separated through "jumps/shortcuts", where the brain dynamics exhibit a jump on the intrinsic manifold from one branch of one state (e.g. N3) to that of another (e.g. awake), as can be seen in Figure S4 and Video S5. Given these results, one can appreciate the effectiveness of the presented intrinsic manifold method to separate different brain states such as different stages of sleep and wakefulness. A detailed explanation of this newly added analysis is included on pages 6 and 10.

Comment 2.6: *p11, "These findings point out the crucial role of nonlinear brain dynamics in order to characterize different brain states" -- this seems to refer to one sense of nonlinearity (of some abstract embedding), but the usual way to test whether brain dynamics are nonlinear is to compare*

to linear surrogate data. e.g., would linear surrogates on the original time series, if fed through the same analysis pipeline, end up with the same result? There are numerous methods for performing such analyses (e.g. Breakspear et al. 2004 HBM, Liegeois et al. 2017 NeuroImage). I note the authors did use permutation testing that shuffled the time series, but this is known to be a strawman surrogate for dynamic functional connectivity, cf. Liegeois et al. 2017.

Answer 2.6: We thank the Reviewer for pointing this crucial piece of information. In the revised version of the manuscript, we use phase randomized surrogates as suggested in Liegeois et al. NeuroImage 2017. The decoding results on the intrinsic manifolds obtained from the original time-series are significantly higher than the results on the linear surrogate manifolds (i.e., on the manifolds obtained from phase-randomized surrogate time-series). In the revised version of the manuscript, we have described this modification in the section Statistical significance analysis, on page 24.

Comment 2.7:

a) The precise workflow from the coherence to the reduced space is a little unclear. So each point in time has a phase coherency matrix (Fig. 1F), and the coherence connectivity dynamics matrix (not shown) captures the similarities between all the phase coherency matrices at all pairs of time points. Then this is pruned into a graph thus linking states across time. There's a mention of a distance matrix in the RMST section, but what is the distance metric here?

Answer 2.7a: We thank the Reviewer for these points. We understand from the Reviewer's comments that the pipeline of our method was not explained in sufficient clarity. The distance matrix in Reviewer's question is constructed from the similarity matrix; i.e. the coherence connectivity dynamics (CCD) matrix, by changing similarities into distances (1-CCD). The minimum spanning tree then performs a minimization over distance weights. We included a detailed explanation of our pipeline into the revised version on page 20.

b) How much of the smoothness of the manifolds is due to imposing some continuity over time?

Answer 2.7b: In our method there is no step that explicitly imposes any temporal continuity. Considering this fact, it is quite remarkable that the method without any explicit temporal constraints yields temporally smooth manifolds within a sleep stage. Yet, it may be worth noting that the low-pass temporal filtering may implicitly impose some degree of temporal continuity. In order to avoid any bias that may be caused by this smoothing, we applied the same filtering to the surrogate time-series.

c) The way that time is encoded in the CCD matrix as all the pairwise comparisons makes it hard for me to intuitively grasp how that summary matrix can be decomposed into distinct states across

time, vs the schematic Fig. 1F-G where I can think of each phase coherency matrix (==time point) occupying a single point in the reduced space.

Answer 2.7c: The Reviewer understanding is correct. Each Phase Coherency matrix corresponds to one time-point, and occupies single time-point in both the high- and the reduced-dimensional spaces. The summary matrix contains all the pair-wise relationships between data points, and represents these relationships in a graph. In fact, the summary matrix corresponds to the adjacency matrix of the graph. Using Laplacian Eigenmaps, we extract the manifold from that graph. The Laplacian eigenfunctions define the basis in which time-points that have similar phase coherency relations are closely located. We have improved the explanation of our pipeline in the revised version on page 5.

Comment 2.8: *How exactly was the PCA calculated? What was the input?*

Answer 2.8: PCA was computed on the pre-processed BOLD time-series, with AAL parcellation and time-points as observations. We used Matlab's *pca()* function. We added this information in the methods in the revised version of the manuscript on page 22.

Comment 2.9: *Did the data include REM sleep? If not, why not? If yes, how were those periods handled.*

Answer 2.9: As indicated throughout the manuscript, the dataset used as input of our method did not include any REM sleep. We feel that the data from the entire sleep cycle including REM sleep would have been very interesting to explore. However our dataset was limited to a sleep cycle of 60 minutes, which did not include REM sleep. The analyzed EEG-fMRI data originates from healthy volunteers who were not sleep deprived. This was assured by a medical interview, sleep diaries and questionnaires. In normal sleep, the first epoch of REM sleep can be expected around 90 min after a subject has fallen asleep. Our subjects in the scanner underwent the 52 min EPI data acquisition right after the necessary preparatory scans. Hence, even if subjects fell asleep immediately in the scanner, the maximum sleeping time could have been 60 mins, and REM sleep is not expected to occur within the scanning time. Indeed, none of the included subjects exhibited REM sleep as verified by sleep scoring of the scanner EEG by sleep experts. This information has been added in the revised version of the manuscript (please see page 18).

Comment 2.10: *The Methods section "Intrinsic manifolds of brain activity" should be in the Results to better guide the reader through what was done (it does a better job of summarizing the workflow than the description of Fig. 1 in the Results).*

Answer 2.10: We thank the Reviewer for pointing this out. We adapted the revised version and included the section “Intrinsic manifolds of brain activity” as the first point in the Results section. We believe this restructuring greatly improved the clarity of our presentation. Please, see page 5.

Comment 2.11: *Figures and supplementary figures are presented together without saying what each shows distinctly (reader needs to work out what is different in the paired supplementary figure).*

Answer 2.11: We thank the Reviewer for pointing this out. We adapted the revised version and included the information on the differences between the figures. Please see pages 7 and 13.

Comment 2.12: *The Atasoy et al. connectome harmonics work is cited as an application of Laplacian eigenmaps, but that was eigenfunctions of the graph Laplacian (those papers never referred to 'eigenmaps'); related to but not the same as the dimensionality reduction technique here. If this is relevant background, then also relevant would be Tokariev et al. 2019 Nature Commun relating cortical eigenmodes to sleep states.*

Answer 2.12: We thank the Reviewer for pointing out the reference of Tokariev et al. Nat. Comm. 2019. We fully agree on its relevance and added it to the revised version of the manuscript. Please see page 4. We believe the citation Atasoy et al. Nat. Comm. 2016 is relevant for this work, as the referred connectome harmonics are estimated as the Laplacian eigenfunctions of brain’s structural connectivity and thus the connectome harmonics framework demonstrates the application of the Laplacian eigenmaps, which are Laplace eigenfunctions, in the spatial domain, i.e. brain’s anatomical connectivity. In this work we utilize the same framework, the Laplacian eigenfunctions, to extract the relevant information from brain dynamics in the temporal domain. We further clarified this link on page 4 in the revised manuscript.

Comment 2.13: *"the intrinsic manifold yields a smooth, continuous representation of the brain dynamics" -- well it's clearly not continuous in the strict sense since there are discrete points, and hence cannot be smooth either.*

Answer 2.13:

We thank the Reviewer for noting this point of possible confusion. The word “continuity”, is used in our manuscript in the spatial sense referring to the structured and non-random spatial distribution of data points on the intrinsic manifold and does not refer to the temporal smoothness, i.e. temporal continuity of the data points on the manifold. We have clarified this point in the revised manuscript on page 6. It may also be worth noting that by definition, Laplacian eigenmaps yield smoothly varying functions:

Intuitively, a function on a graph is smooth if its value at a node is similar to its value at each of the node's neighbors (<https://arxiv.org/abs/2011.01307>, Section 4.1.) Mathematically, the smoothness of a function can be measured by the Dirichlet energy defined as:

$$E(\mathbf{f}) = \int_{\mathcal{M}} \|\nabla \mathbf{f}(x)\|^2 dx = \int_{\mathcal{M}} \mathbf{f}(x) \Delta \mathbf{f}(x) dx$$

Since the Laplace operator (Δ) may be thought of as the functional derivative of the Dirichlet energy, Laplace eigenfunctions (Laplacian Eigenmaps) are estimated by maximizing the smoothness of the embedding. For a detailed explanation on the smoothness of Laplacian eigenfunctions, please refer to <https://arxiv.org/abs/2011.01307>, section 4.1. Also, in the seminal paper of *Belkin and Niyogi Neural Computation 2003*, which introduces the method of Laplacian Eigenmaps used in our method, the idea of maximizing smoothness is expressed in terms of minimizing “how far apart \mathbf{f} maps nearby points”.

Comment 2.14: *Fig. 3, there are two panel Es, and in the caption "(D) and H)" is meant to be "(G) and H)" (after fixing the E duplication).*

Answer 2.14: We thank the Reviewer for pointing this out. These typos have been corrected in the revised version. Please see page 12.

Comment 2.15: *p8, accuracy of "68±1%" seems peculiarly precise, how is the uncertainty calculated.*

Answer 2.15: This accuracy corresponds to the average across the mean accuracies in awake stage comparisons. Its uncertainty is calculated as follows:

$$e_{Awake} = \frac{1}{3} \sqrt{e_{A-N1}^2 + e_{A-N2}^2 + e_{A-N3}^2},$$

where e_{A-x}^2 is the average classification accuracy uncertainty for each awake stage comparison (standard deviation across subjects of the mean cross-validation accuracy). This is stated in the revised manuscript on page 23.

Comment 2.16: *"PCA was unable to capture the structure underlying brain dynamics" -- well, it was weakly but significantly different from chance so it partly captures something (PCA clearly performed poorly, but don't overcook it).*

Answer 2.16: We thank the Reviewer for pointing this out. In the revised version, we changed “unable to capture” for “performed poorly at capturing”. Please see page 10.

Comment 2.17: *Does PCA's failure come down to nonlinearity or something else? Just because a NL method does well and a linear method does not, doesn't mean that the fault lies with the linearity.*

Answer 2.17: We thank the Reviewer for pointing this idea out. We understand the Reviewer's concern when pointing out that the low performance of PCA (in comparison to intrinsic manifold method) in the sleep stage classification task can be attributed to PCA's linear nature or to other details of its framework. Inspired by the Reviewer's comment, and in order to fully assess the role of the nonlinearities in the superior performance of our method, we compared the intrinsic manifold not only to PCA but also to linear surrogate manifolds. These linear surrogate manifolds were estimated similarly to the intrinsic manifolds, with the only difference being that the phase of the time-series was randomized (see Methods). These results revealed that we can be certain that nonlinearities play a crucial role in the data, as indicated by the significant improved accuracies on the intrinsic manifolds with respect to the linear surrogate manifolds. This explanation has been included in the revised version of the manuscript (see page 12)

Comment 2.18: *It is mentioned that this classifier outperforms previous efforts to detect sleep states from fMRI, but by how much? This should be quantified.*

Answer 2.18: We thank the Reviewer for this important comment. We have addressed this by including a further comparison to previous methods, as also asked for by Reviewer 1. Please refer to our response to the comments R1.1 and R1.2.

Comment 2.19: *Regarding dynamics existing on subspaces of the high-dimensional phase space, this idea of course goes back a long way, at the very least to Haken, and obviously all the work on modeling neural masses and fields etc. And even ignoring such theoretical efforts, fMRI obviously has spatial smoothness so no one should think of every voxel being independent.*

Answer 2.19: We thank the Reviewer for pointing out the overlap between our work and the suggested literature. We have included citations referring to Herman Haken and following related work on metastability and quasi-attractor dynamics in the Introduction of the revised manuscript. We have also clarified the fact that voxels in fMRI data are not independent due to spatial smoothness (please see page 2).

Comment 2.20: *Abstract mentions "nonlinear transitions", but "transitions" doesn't appear again. In what sense are the transitions nonlinear? Also not clear that transitions are even studied here.*

Answer 2.20: We thank the Reviewer for pointing this out. Indeed, this work does not explicitly study transitions, but only the differences between sleep stages in terms of brain dynamics, although

the sleep stage transitions correspond to jumps in the trajectory of brain dynamics on the intrinsic manifold, as shown in Supplementary Figure S4, and as also discussed in our response to comment R2.5. To improve the clarity of our presentation, following the Reviewer's comment, we have removed the only mention of the term "transitions" (which was located in the abstract).

Comment 2.21: *The Methods presents the Hilbert transform to a peculiar depth; it could be shortened. Did the authors just use MATLAB's hilb()? If not, why not?*

Answer 2.21: We thank the Reviewer for pointing this out. We removed the detailed explanation of the Hilbert transform in this revision, and included a reference to the MATLAB's function has been added in the Coherence Connectivity Dynamics section. Please see page 21.

Comment 2.22: *"We first extracted the instantaneous phase signal of the ultraslow fluctuations (0.04 - 0.07 Hz)" vs "Data were band-pass filtered in the range 0.01–0.1 Hz" -- which was it? And why 0.04-0.07 Hz anyway?*

Answer 2.22: We thank the Reviewer for pointing this out. In the revised version of the manuscript, we used a band-pass filter in the range 0.04 - 0.07 Hz. The choice of this frequency range was driven to the findings of *Glerean et al. Brain Connectivity 2012*, which points out the importance of using this particular frequency range for bandpass filtering when using the Hilbert transform to estimate functional connectivity based on phase coherency.

Comment 2.23: *p18, typo "RMTS"*

Answer 2.23: We thank the Reviewer for pointing this out. The typo is corrected in this revision.

Comment 2.24: *Why does the dimensionality reduction section review other methods? If including that material, it should be earlier in the paper to provide more context, or else just remove it to keep things simpler.*

Answer 2.24: We thank the Reviewer for pointing out that we overly detailed the section on dimensionality reduction methods. We have shortened this part in the revised version of the manuscript, focusing only on the techniques used in this work, namely Principal Component Analysis and Laplacian Eigenmaps. We have taken the liberty of preserving some key details on dimensionality reduction that we found crucial to understand the similarities and differences between the two aforementioned techniques.

RESPONSE TO REVIEWER 3

Comment 3.0: *The current study presents a novel method for calculating low dimensional manifolds in fMRI data and applies it to data at different sleep/wake cycles. I thought the paper*

was interesting considering recent trends to investigate low dimensional manifolds in fMRI data. In general the paper was well written with clear aims etc., I have a few comments below.

Answer 3.0: We thank the Reviewer for their constructive and supportive comments. We believe that through their and other Reviewers' suggestions, the presentation and quality of our paper has further improved in the revised manuscript.

Comment 3.1: *Was any further fMRI preprocessing performed? The methods seem very sparse compared to common practice where nuisance variables such as head motion are regressed (Ciric et al., 2017).*

Answer 3.1: We thank the Reviewer for noticing the lack of preprocessing steps described. In the revised version manuscript, we have added the complete fMRI pre-processing pipeline, including the description of noise and motion regression and provided a complete description of every preprocessing step. Please refer to fMRI pre-processing subsection of Methods, on page 19.

Comment 3.2: *Projecting the dimensions back into brain space would be helpful for the reader to contextualize the manifolds - which are by nature quite abstract - e.g., what the authors have done in Figure 1E.*

Answer 3.2: We thank the Reviewer for this comment. We would like to point out that due to the nature of the nonlinear dimensionality reduction, the mapping function from the high- to the low-dimensional space is not available. Therefore, it cannot be reversed to project the data-points back into the high-dimensional space. Unlike linear methods like PCA, these methods reveal directly the low dimensional coordinates of the data points, and the nonlinear mapping is implicitly applied in the estimation of these coordinates but the mapping function is not explicitly available as it is in linear methods such as PCA. For this reason, in Figure 1E we illustrate how phase coherence changes along the manifold to illustrate the steps our method takes to estimate the intrinsic manifold from the high dimensional fMRI data. We have described this aspect of our method in the revised version of the manuscript (see page 22).

Comment 3.3: *The paper has a few typos, e.g., page 21 paragraphs regarding SVM, 'splitting' should probably be 'split', 'resting' should probably be 'remaining'.).*

Answer 3.3: We thank the Reviewer for pointing this out. The typos are corrected in this revision.

Comment 3.4: *I am wondering why the predictions were on a stage-by-stage basis and not all sleep categories in a single classification model. Wouldn't an 'all in one' analysis demonstrate the point in a simpler fashion? (which could have the current analyses as follow-ups)*

Answer 3.4: We thank the Reviewer for this important point. We agree with the Reviewer that an additional all in one sleep stage type of classification would be adequate in order to be able to demonstrate brain decoding in a simpler fashion. In the revised version of the manuscript, we have included this additional “all in one” classification analysis, explained in detail in methods (see page 23) and reported the results of decoding on page 7.

Please also refer to our comment to Reviewer 2.3.

Comment 3.5: *Relatedly, is it possible that different features contributed to the classification accuracy depending on the given contrast? Is it possible to, for example, plot the SVM weights to see which features contributed?*

Answer 3.5: We thank the Reviewer for this comment. We have thought about exploiting this information. The features considered (Eigenvectors of the CCD graph Laplacian) indicate gradients of low variability among temporal points. These features do not explicitly contain spatial information about brain maps, but only temporal relationships among data-points. After observing the low-dimensionality of the intrinsic manifold, it seemed only 7 dimensions contribute significantly in the decoding. This contribution is further explained by the dimensionality analysis in Figure 3L,N.

Comment 3.6: *The authors compare the current method to PCA - another method used in fMRI to identify low dimensional states. Would a more appropriate ‘null’ model be data that is high dimensional?*

Answer 3.6: We thank the Reviewer for this important comment. We agree that a high-dimensional space is a very appropriate null model to test that data is low-dimensional. Following the Reviewer’s suggestion, in the revised version of the manuscript, we now use PCA as a null model for linearity (in the sleep stage differences), and high dimensional data (with 90 parcels AAL parcellation the original dimensionality of the data is 90). Specifically, in the revised version, we tested whether the intrinsic manifolds were low dimensional by comparing the classification accuracies in manifolds of increasing dimensionality, and by comparing them against the classification in the original high dimensional data. Our results indicate that from $d=7$ to the original dimensionality (90), adding new dimensions in the intrinsic manifold does not improve the classification accuracy (for all dimensions $d>7$, $p\text{-value}>0.05$, Wilcoxon Rank-sum two-sided test, corrected for multiple comparisons via FDR). These results indicate that the different sleep stages can be optimally decoded from the intrinsic manifolds with dimensionality $d=7$. A detailed explanation has been included in the revised version of the manuscript (see page 8).

Reviewers' Comments:

Reviewer #1:

Remarks to the Author:

No more questions or suggestions from me.

Congratulations to the authors.

Reviewer #2:

Remarks to the Author:

The authors have done a good job in revising the paper.

Reviewer #3:

Remarks to the Author:

The authors have adequately addressed all of my concerns. Congratulations on a nice paper.